JCB Journal of Cell Biology

# REPORT

# Peroxisomal ROS control cytosolic *Mycobacterium tuberculosis* replication in human macrophages

Enrica Pellegrino[1], Beren Aylan[1], Claudio Bussi[1], Antony Fearns[1], Elliott M. Bernard[1], Natalia Athanasiadi[1], Pierre Santucci[1], Laure Botella[1], and Maximiliano G. Gutierrez[1]

**Peroxisomes are organelles involved in many metabolic processes including lipid metabolism, reactive oxygen species (ROS) turnover, and antimicrobial immune responses. However, the cellular mechanisms by which peroxisomes contribute to bacterial elimination in macrophages remain elusive. Here, we investigated peroxisome function in iPSC-derived human macrophages (iPSDM) during infection with *Mycobacterium tuberculosis* (Mtb). We discovered that Mtb-triggered peroxisome biogenesis requires the ESX-1 type 7 secretion system, critical for cytosolic access. iPSDM lacking peroxisomes were permissive to Mtb wild-type (WT) replication but were able to restrict an Mtb mutant missing functional ESX-1, suggesting a role for peroxisomes in the control of cytosolic but not phagosomal Mtb. Using genetically encoded localization-dependent ROS probes, we found peroxisomes increased ROS levels during Mtb WT infection. Thus, human macrophages respond to the infection by increasing peroxisomes that generate ROS primarily to restrict cytosolic Mtb. Our data uncover a peroxisome-controlled, ROS-mediated mechanism that contributes to the restriction of cytosolic bacteria.**

## Introduction

Peroxisomes are dynamic, single membrane-bound organelles that regulate important metabolic processes such as fatty acid β-oxidation, biosynthesis of ether phospholipids, and metabolism of reactive oxygen species (ROS; Wanders et al., 2023). Peroxisomes have been shown to be important in innate immunity (Di Cara, 2020), antiviral signaling (Ferreira et al., 2022), phagocytosis of bacteria (Di Cara et al., 2018), and activating the expression of Type III interferons during infection (Odendall et al., 2014; Dixit et al., 2010). Peroxisomes can be either a source or a sink of ROS to regulate oxidative stress and cellular homeostasis (Fransen et al., 2012). In fact, peroxisomes are major producers and scavengers of ROS, including hydrogen peroxide ($H_2O_2$) generated as a byproduct during β-oxidation (Schrader and Fahimi 2006). Although $H_2O_2$ is involved in inter/intracellular signaling during antibacterial host defense (Behera et al., 2022; Bedard and Krause 2007; Blander and Sander 2012), little is known about the role of peroxisomal $H_2O_2$ in this process.

ROS generation during phagosome maturation is a crucial step to restrict intracellular pathogen proliferation in macrophages (Flannagan et al., 2012). ROS production is mostly restricted to phagosomes either by the nicotinamide adenine dinucleotide phosphate (NADPH) oxidase (Mantegazza et al., 2008) or the mitochondria (Geng et al., 2015), but whether there is a source of ROS in the cytosol that can restrict bacteria is unknown. Targeting bacteria into phagosomes facilitates recruitment of lysosomal enzymes and the local generation of toxic ROS through the recruitment of the NADPH oxidase Nox2 isoform (Gluschko et al., 2018; Pollock et al., 1995). However, many intracellular bacteria, such as *Mycobacterium tuberculosis* (Mtb), damage the phagosome to access the more permissive cytosol (van der Wel et al., 2007; Lerner et al., 2017). In the cytosol, bacteria are recognized by the autophagy machinery and targeted to a membrane-bound compartment where they can be redirected to the lysosomal pathway for removal (Watson et al., 2015; Bussi and Gutierrez, 2019). These antibacterial mechanisms operating in the cytosol target the bacteria to membrane-bound compartments (Bernard et al., 2020); however, less is known about direct mechanisms restricting cytosolic mycobacteria.

Here, we combined human stem cell–derived macrophages (iPSDM; Pellegrino and Gutierrez 2021) with CRISPR/Cas9

[1]Host-pathogen interactions in Tuberculosis Laboratory, The Francis Crick Institute, London, UK.

Correspondence to Maximiliano G. Gutierrez: max.g@crick.ac.uk

E.M. Bernard's current affiliation is Department of Immunobiology, University of Lausanne, Epalinges, Switzerland.   P. Santucci's current affiliation is Aix-Marseille Université, Centre national de la recherche scientifique, Laboratoire d'Ingénierie des Systèmes Macromoléculaires, Institut de Microbiologie de la Méditerranée FR3479, Marseille, France.

genome editing to deplete peroxisomes and novel genetically encoded reporters to monitor ROS localization during Mtb infection. Through this approach, we demonstrate a localization-dependent action of peroxisomal ROS that restricts the replication of cytosolic bacteria. Our data uncover two independent ROS antibacterial mechanisms: one involving the NADPH oxidase operating on phagosomes and the second a peroxisomal ROS-dependent antibacterial mechanism that operates in the cytosol of human macrophages.

## Results and discussion

### Mtb infection induces ESX-1–dependent peroxisome biogenesis in human macrophages

To study the role of peroxisomes during Mtb infection in human macrophages, we infected human iPSDM with Mtb WT or a mutant that lacks a functional ESX-1 T7SS (Mtb ΔRD1; Bernard et al., 2020). An RNA sequencing analysis showed differential expression of genes involved in peroxisome fission and biogenesis after 48 h of infection with Mtb WT but not Mtb ΔRD1 (Bernard et al., 2020; Fig. 1 A). To investigate this RD1-dependent phenotype, we analyzed the biogenesis and dynamics of peroxisomes by imaging peroxisomal proteins. PEX14 is a peroxisomal membrane protein essential to generate mature, functional peroxisomes and an early marker of peroxisome formation (Sugiura et al., 2017; Albertini et al., 1997). Peroxisomal matrix proteins are synthesized in the cytoplasm and imported after translation across the peroxisome membrane by the recognition of a specific peroxisomal targeting signal (PTS) at the N-terminus (PTS2) or C-terminus (PTS1; Aitchison et al., 2013). Thus, to distinguish new peroxisomes from mature peroxisomes, we generated induced pluripotent stem cells (iPSC) expressing enhanced green fluorescent protein (EGFP) fused with PTS1 at the safe-harbor locus AAVS1 to label mature peroxisomes (Fig. 1 B). A 3D analysis of the number of PEX14$^+$ peroxisomes showed an increase in the number of PEX14$^+$ peroxisomes per cell in iPSDM infected with Mtb WT, but not Mtb ΔRD1, at 24 h but not 48 h after infection (Fig. 1, C and E). Western blot analysis of the peroxisomal proteins PEX14, ACOX1, HSD17B4, and Catalase (CAT) confirmed an increase in total peroxisomal proteins after 24 h of infection with Mtb WT but not Mtb ΔRD1 (Fig. 1 I). However, infection with Mtb WT did not result in significant differences in the number of EGFP-PTS1$^+$ peroxisomes (Fig. 1 F, pipeline in Fig. 1 D). When we analyzed the marker EGFP-PTS1 and PEX14, we observed that the majority of PEX14-positive peroxisomes induced by Mtb WT infection were negative for EGFP-PTS1 by colocalization analysis (Fig. 1 J). This increase in PEX14$^+$ peroxisomes was not significant after 48 h of infection (Fig. 1 E) Moreover, there was no correlation between Mtb WT burden and the peroxisomal content at the single-cell level after 24 and 48 h of infection (Fig. 1 G), suggesting that bacterial replication was not linked to the increase in PEX14-positive peroxisomes. We next analyzed whether Mtb affects peroxisome morphology, which is normally associated with a change in peroxisome function (Ribeiro et al., 2012). Quantitative analysis of peroxisome morphology showed that peroxisomes were more elongated after 24 h of infection

with Mtb WT but not with Mtb ΔRD1, suggesting changes in peroxisome function due to selective fission and/or fusion events (Fig. 1, C and H). Altogether, in human macrophages, Mtb infection triggers an increase in the number of peroxisomes and induces changes in peroxisome dynamics in an ESX-1–dependent manner.

### Peroxisomal activity is required to restrict Mtb in human macrophages

To further understand if this increase in peroxisome number was a response of macrophages to control Mtb WT, we generated iPSC knockout (KO) for the peroxisome biogenesis factor PEX3 (Fig. S1, A and B). Two independent edited iPSC clones were tested for pluripotency-associated markers: PEX3$^{-/-}$ iPSC clones were positive for OCT3/4 and expressed TRA-1-60 and TRA-1-81 (Fig. S1 C). iPSC were differentiated into iPSDM and characterized by flow cytometry; the deletion of PEX3 did not significantly impact macrophage differentiation (Fig. S1 D). PEX3$^{-/-}$ iPSDM lacked peroxisomes with GFP-PTS1 remaining cytosolic in contrast to PEX3$^{+/+}$ iPSDM, and this phenotype was rescued after expression of PEX3 gene using the PEX3-Turbo plasmid (Fig. 2 A and Fig. S2 A). iPSDM lacking PEX3 showed decreased expression of peroxisomal genes, such as ABCD1, PEX11b, PEX19, and PEX14. In addition, PEX3$^{-/-}$ were not able to process and activate ACOX1 and HSD17B4 (Fig. S2 B; Osumi et al., 1980; Kurochkin et al., 2007). Interestingly, in the absence of peroxisomes, we detected an increased expression of CAT, an important enzyme for $H_2O_2$ degradation (Fig. S2 C). Moreover, in cells lacking peroxisomes, CAT mislocalized in the nucleus and cytosol of cells lacking peroxisomes, and PEX14 localized in TOM20$^+$ structures, suggesting a relocalization into mitochondria as previously reported (Sugiura et al., 2017; Fig. 2 A). When PEX3$^{-/-}$ macrophages were infected with Mtb WT, we observed an increase of Mtb replication, calculated as growth index (mean Mtb area per cell 72 hpi – mean Mtb area per cell 2 hpi)/(mean Mtb area per cell 2 hpi), in PEX3$^{-/-}$ up to 72 h after infection when compared with PEX3$^{+/+}$ macrophages (Fig. 2 B). In contrast, Mtb ΔRD1 growth was similar between PEX3$^{+/+}$ and PEX3$^{-/-}$ iPSDM after 72 h of infection, indicating that PEX3 was only required for the control of Mtb WT (Fig. 2 C). Different from previous reports (Di Cara et al., 2018), we observed no differences in Mtb WT or Mtb ΔRD1 phagocytic uptake at 2 h after infection between WT and PEX3 KO iPSDM (Fig. S2 D). Altogether, these data show that peroxisomal function is required for the control of Mtb WT but not Mtb ΔRD1, suggesting that peroxisomes are required to control bacteria that are able to reach the cytosol.

### Peroxisomal ROS increase after infection with Mtb WT

We next investigated how increased peroxisome biogenesis and altered morphology restrict cytosolic Mtb WT replication. Initially, it was thought peroxisomes were only responsible for the decomposition of $H_2O_2$ via CAT; however, it is now clear that peroxisomes can contribute to the generation of $H_2O_2$ through a variety of enzyme reactions, such as the β-oxidation of fatty acids. In cultured cells, oxidative stress has been shown to induce morphological changes of the peroxisomal compartment

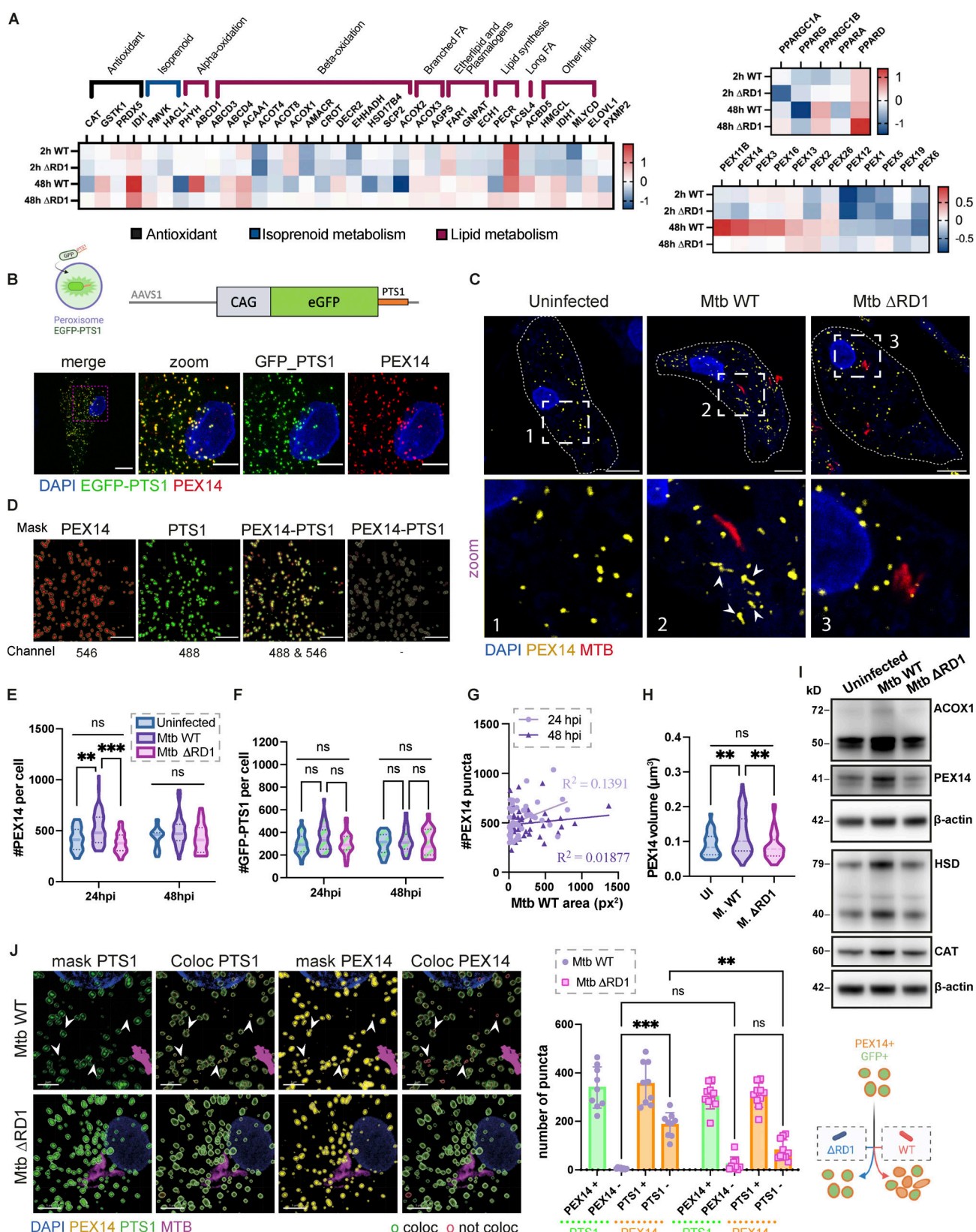

**Figure 1. Mtb infection triggers peroxisome biogenesis in human macrophages. (A)** Heatmap of differentially expressed genes from RNA sequencing analysis of iPSDM infected with either Mtb WT or Mtb ΔRD1 for 2 or 48 h. Data normalized to the uninfected control. Left: RNA expression of peroxisomal protein associated with β-oxidation, α-oxidation, transport of lipid, and antioxidant activity. Top right: PPAR genes regulation during infection. Bottom right: RNA expression of peroxisomal protein associated with membrane assembly, cargo, and division of peroxisomes. **(B)** Top: Schematic representation of the CMV:EGFP-PTS1 allele in the AAVS1 locus. Bottom: Confocal images of iPSDM expressing EGFP-PTS1 (green) stained for PEX14 (red) and nuclear staining

(blue). Scale bars: 10 µm, zoom: 5 µm. **(C)** Confocal images of iPSDM after 24 h of infection (Uninfected, Mtb WT and ΔRD1). Nuclear staining (blue), PEX14 staining (orange), and Mtb-E2-Crimson (red). Scale bars: 10 µm. **(D)** Pipeline for masking PEX14 (red) and EGFP-PTS1 (green) to perform colocalization study. Arrowheads point at elongated peroxisomes. Scale bars: 5 µm. **(E)** Analysis of peroxisome number during Mtb infection. Violin plot displaying the distribution of PEX14 puncta; quantification from $N > 80$ cells analyzed at each time point. Data were collected from three independent experiments. Quantiles representing the distribution are shown. **(F)** Quantification of EGFP-PTS1 puncta per cells. $N > 80$ cells were quantified for each time point from three independent experiments. **(G)** Pearson correlation between Mtb area and PEX14 puncta in infected iPSDM with Mtb WT and ΔRD1, at 24 hpi (light violet) and 48 hpi (dark violet). Results are from 50 cells obtained from three independent experiments. **(H)** Quantification of PEX14 volume ($µm^3$) per cell (UI = uninfected, M. WT = Mtb WT, M. ΔRD1 = Mtb ΔRD1.) $N > 80$ cells were quantified from three independent experiments. **(I)** Western blot analysis of peroxisomal protein PEX14, ACOX1, CAT, and HSD17B4 (HSD) of iPSDM at 24 hpi of infection (Uninfected, Mtb WT, and ΔRD1). **(J)** Analysis of colocalization of PEX14 and EGFP_PTS1 peroxisomes. Left: Confocal images of iPSDM at 24 hpi infected with Mtb WT and ΔRD1. Nuclear staining (blue), PEX14 staining (orange), EGFP_PTS1 (green), and Mtb-E2-Crimson (magenta). Arrowheads point at PEX14⁺/PTS1⁻ peroxisomes. Scale bars: 10 µm. Right: Quantification of PEX14/PTS1 colocalization per cells. Error bars indicate SD. Data representative from one out of three independent experiments. $N = 60$ cells were quantified. Significance was determined by two-way ANOVA with Tukey's multiple comparison post-test (C, E, F, and J). P value 0.002 (**), <0.0001 (***). Source data are available for this figure: SourceData F1.

(Schrader and Fahimi 2006), and it is known that ROS are a major antibacterial defense mechanism used by macrophages upon activation (Herb and Schramm 2021). As a major fraction of Mtb WT is in the cytosol of iPSDM at 48 h after infection in our infection model (Bernard et al., 2020), we hypothesized that the peroxisome-dependent antimycobacterial effect was due to ROS turnover in the cytosol. Given that the commercially available probes lacked subcellular spatial resolution, we decided to express genetically encoded fluorescent sensors to monitor ROS levels in live cells. We generated iPSC expressing the $H_2O_2$-sensitive HyPer protein (Belousov et al., 2006) targeted either to the cytosol (Cyto_Hyper), peroxisomes (Pexo_Hyper), or endosomes (Endo_Hyper) to define the oxidative states in the cytosol, peroxisomes, or endolysosomal compartment, respectively (Fig. S3, A–C). We infected the Pexo_Hyper iPSDM with Mtb WT and Mtb ΔRD1 and monitored $H_2O_2$ levels by live cell imaging for 30 h. By using a high-content, single-cell analysis imaging approach, we detected an increase of peroxisomal $H_2O_2$ only in macrophages infected with Mtb WT and not in iPSDM infected with Mtb ΔRD1 or in the uninfected cells (Fig. 3, A and B). This increase in ROS production was observed in iPSDM infected with Mtb WT but not in the bystander cells, indicating that the presence of intracellular bacilli triggered peroxisomal ROS formation (Fig. 3 C). We hypothesize that this ROS generation could lead to a reduction of peroxisomes at 48 hpi (Fig. 1 A) due to the accumulation of oxidative damage within the peroxisomes. Altogether, our data show that ROS are generated in peroxisomes after infection with Mtb in human macrophages (Fig. 3, A–C), suggesting that peroxisomes are implicated in controlling cytosolic bacteria by regulating redox homeostasis.

### PPARα activation restricts Mtb replication by inducing peroxisomal ROS

An increase of peroxisomal ROS has been linked to either an increase of peroxisomal β-oxidation or the inhibition of CAT activity (Schrader and Fahimi 2006; Ruiz-Ojeda et al., 2016). In our Western blot analysis, we observed a significant increase of two peroxisomal proteins: ACOX1 and HSD17B4, indicating a potential upregulation of β-oxidation and peroxisomal ROS during Mtb infection (Fig. 1 I). Thus, we pharmacologically modulated the function of peroxisomes with sodium 4-phenylbutyrate (4-PBA), which increases peroxisomal biogenesis and induces β-oxidation

(Kemp et al., 1998; Jean Beltran et al., 2018), GW7647 (GW), which induces β-oxidation and peroxisome proliferation in a PPARα-dependent manner (McMullen et al., 2014; Schrader et al., 2016), and 3-amino-1,2,4-triazole (3-AT), which inhibits catalase activity (Ueda et al., 2003). In contrast to 3-AT and GW, we found that 4-PBA strongly reduced Mtb growth in vitro (Fig. S3, D–F) and it was not subsequently tested. We observed a pronounced increase in the expression of peroxisomal genes involved in fission, biogenesis, and β-oxidation in iPSDM after treatment with GW compared with untreated and 3-AT–treated iPSDM (Fig. S3 G). This increase in gene expression was not associated with an increase in peroxisomal number or changes in morphology in the GW-treated cells (Fig. 3, D and E). Next, we monitored peroxisomal $H_2O_2$ production with the Pexo_Hyper reporter in iPSDM treated with either GW or 3-AT. After 24 h of treatment, we detected an increase in peroxisomal ROS (Fig. 3 F), mostly with GW. Finally, iPSDM were treated with the modulators, infected with Mtb WT or Mtb ΔRD1, and bacterial replication was analyzed by high-content imaging (Fig. 3 G). 3-AT had no effect on bacterial replication in iPSDM after 48 h of infection. Conversely, GW significantly restricted Mtb replication in iPSDM, suggesting that PPARα-mediated β-oxidation and peroxisomal ROS generation contribute to Mtb restriction by human macrophages (Fig. 3 G). Strikingly, this effect was observed after infection with Mtb WT and there was no effect at later time points of any of the compounds tested in macrophages infected with Mtb ΔRD1 (Fig. 3 G). These results indicate that peroxisome dynamics and the increase in peroxisomal ROS observed after infection contribute to restricting Mtb WT but not Mtb ΔRD1, arguing for a localization-dependent action of peroxisomal ROS.

### Peroxisome-dependent restriction of Mtb is associated with higher levels of ROS in the cytosol

$H_2O_2$ is a permeable and diffusible molecule involved in inter- and intracellular signaling during host defense (Stone and Yang 2006). To test if the increase in peroxisomal ROS in Mtb WT infection (Fig. 3, A–C) was caused by peroxisomal activity regulation rather than ROS internalization (e.g., sink effect), we monitored cytosolic ROS in infected macrophages with and without peroxisomes. We infected PEX3⁺/⁺ and PEX3⁻/⁻ iPSDM, expressing the Cyto_Hyper reporter, with Mtb WT and Mtb ΔRD1 and analyzed $H_2O_2$ by live cell imaging (Fig. 4, A–C). At 40

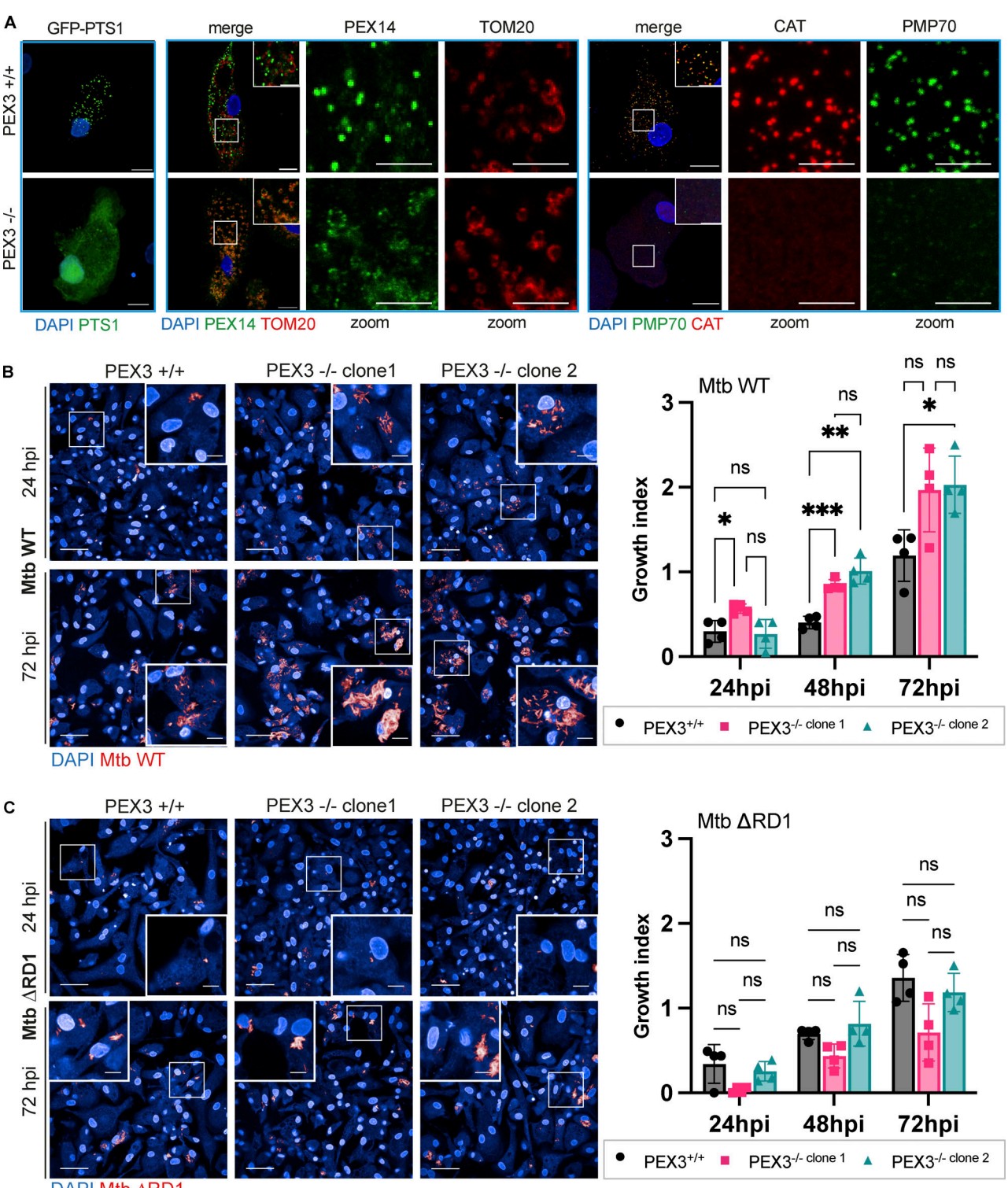

Figure 2. **Human macrophages lacking peroxisomes are unable to restrict Mtb WT replication. (A)** Confocal images of iPSDM PEX3[+/+] and PEX3[−/−] for GFP-PTS1 (green), PEX14 (green), TOM20 (red), CAT (red), and PMP70 (green). Nuclear staining (blue). Scale bars: 10 μm. **(B)** Analysis of Mtb WT growth in iPSDM lacking peroxisomes. Left: Confocal images of iPSDM PEX3[+/+] and PEX3[−/−] (clone 1 and 2) at 24 and 72 hpi infected with Mtb WT. Nuclear staining (blue) and Mtb-E2-Crimson (red). Scale bars: 100 μm, zoom: 10 μm. Right: Growth index of Mtb WT in iPSDM control (PEX3[+/+]) or KO for PEX3 (PEX3[−/−] clones 1 and 2). Data representative of one out of three independent experiments ($n$ = 4 independent wells). **(C)** Analysis of Mtb ΔRD1 growth in iPSDM lacking peroxisomes. Left: Confocal images of iPSDM PEX3[+/+] and PEX3[−/−] (clone 1 and 2) at 24 and 72 hpi infected with Mtb ΔRD1. Nuclear staining (blue) and Mtb-E2-Crimson (red). Scale bars: 100 μm, zoom 10 μm. Right: Growth index of Mtb WT in iPSDM control (PEX3[+/+]) or KO for PEX3 (PEX3[−/−] clones 1 and 2). Data are representative of one out of three independent experiments ($n$ = 4 technical replicate per each condition). Significance was determined by two-way ANOVA with Tukey's multiple comparison post-test (B and C). P value 0.033 (*), 0.002 (**), <0.0001 (***).

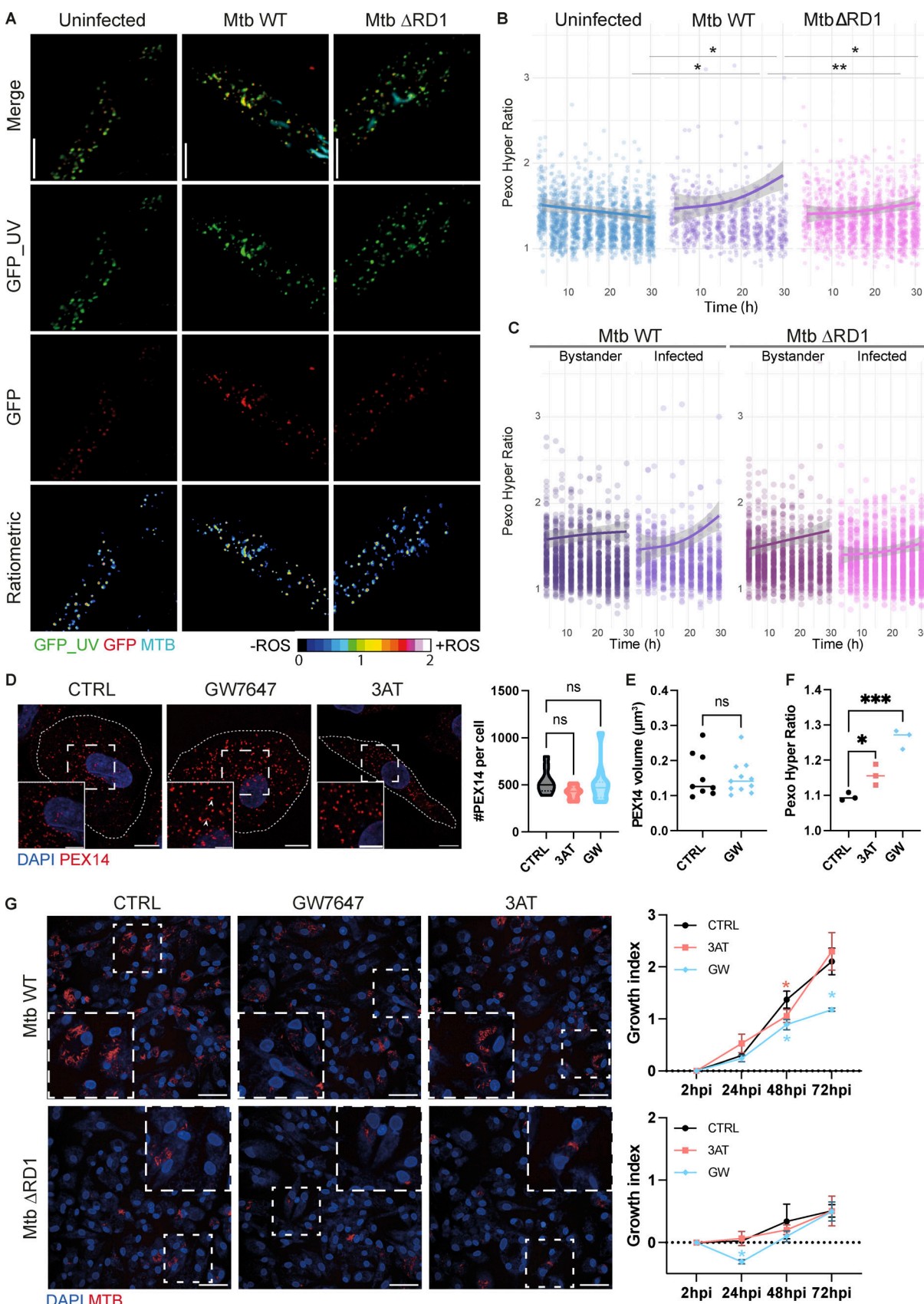

Figure 3. **Peroxisomal H₂O₂ increases after infection with Mtb WT and it is important to control cytosolic replication in macrophages. (A–C)** Analysis of Pexo_Hyper ratio in iPSDM during infection (Uninfected, Mtb WT, and ΔRD1). **(A)** Snapshot of Pexo_Hyper reporter during Mtb infection. Confocal images of iPSDM at 24 hpi of infection. GFP_UV (green), GFP (red), and Mtb-E2-Crimson (cyan) and a ratiometric imaging of the Pexo_Hyper reporter. Scale bars: 10 μm.

**(B)** Each point represents the ratio of cells infected over 30 h along with a trend line and standard deviation. A total of more $N > 300$ cells were quantified at each time point. Significance was determined for the last two time points (27 and 30 hpi) by two-way ANOVA with Tukey's post-doc test. P value (APA) 0.033 (*), 0.002 (**), <0.0001 (***). **(C)** The left graph shows the Pexo_Hyper ratio of iPSDM infected and bystander in the Mtb WT infected well. The right graph shows the ratio of iPSDM infected and bystander in the Mtb ΔRD1 infected well. $N > 300$ cells were quantified per each time point. **(D)** Analysis of peroxisome's number with peroxisome modulators. Left: Confocal imaging of iPSDM treated for 48 h with GW or 3-AT. Nuclear staining (blue) and PEX14 (red). Scale bars: 10 μm, zoom: 5 μm. Right: Quantification of PEX14 puncta per iPSDM control (CTRL) or treated with the drug (GW and 3-AT). Data representative from one out of two independent experiments. $N = 30$ cells were quantified. **(E)** Quantification of PEX14 volume ($μm^3$) per cells. Data representative from one out of two independent experiments. $N = 20$ cells were quantified. **(F)** Quantification of Pexo_Hyper ratio in iPSDM untreated (CTRL) or treated with the drug (GW). Data are representative of one out of two independent experiments ($n = 3$ independent wells per replicate), $N > 200$ cells were quantified. **(G)** Analysis of Mtb growth with peroxisome modulators. Left: Confocal images of iPSDM at 72 hpi infected with Mtb WT (top) and ΔRD1 (bottom). Nuclear staining (blue) and Mtb-E2-Crimson (red). Scale bars: 100 μm. Right: Growth index of Mtb WT (top) and Mtb ΔRD1 (bottom), for iPSDM control (CTRL) or treated with the drug (GW and 3-AT). Data representative from one out of three independent experiments (bars represent SD of $n = 3$ independent wells per replicate). Significance was determined using unpaired $t$ test (E), one-way ANOVA with Dunnett's multiple comparison post-test (D and F), and by two-way ANOVA with Dunnett's multiple comparison post-test (G) and Tukey's multiple comparisons test (B). P value 0.033 (*), 0.002 (**), and <0.0001 (***).

hpi, we detected an increase in cytosolic $H_2O_2$ in macrophages PEX3$^{+/+}$ infected with Mtb WT and Mtb ΔRD1, although this was not significant (Fig. 4, A and B). Interestingly, in PEX3$^{-/-}$ iPSDM, $H_2O_2$ production in the cytosol dropped after Mtb WT infection (Fig. 4, A and B). The fluorescent intensity profiles along the cytosolic Mtb were determined, confirming that $H_2O_2$ was decreasing in iPSDM lacking peroxisomes infected with Mtb WT (Fig. 4, B and C). By using the Endo_Hyper reporter, we monitored phagosomal ROS in an area of the bacteria expanded by 0.5 μm in all directions to capture the proximal environment surrounding the bacteria (Fig. 4, D and E). We found a decrease in phagosomal ROS with Mtb WT in PEX3$^{-/-}$ iPSDM but not with Mtb ΔRD1 infection, suggesting that Mtb WT accessing the cytosol were facing a less oxidative environment in PEX3$^{-/-}$ iPSDM (Fig. 4 E). Altogether, our ROS-localization analysis in human macrophages showed a peroxisome-dependent source and action of $H_2O_2$ to restrict cytosolic Mtb.

**Peroxisome-dependent restriction of Mtb is spatiotemporally distinct from NADPH oxidase activity**

The NADPH oxidase Nox2 isoform is the main source of ROS, which are a crucial component of the antimicrobial activity of professional phagocytes, including macrophages (Gluschko et al., 2018; Pollock et al., 1995). However, Nox2 is normally assembled onto membranes and generates ROS within the lumen of vesicles (Herb and Schramm 2021). We found that most inhibitors currently used to inhibit ROS, such as DPI, affected macrophage viability (data not shown) and directly inhibited Mtb growth in vitro, as previously shown (Nguyen et al., 2018; Yeware et al., 2019; Pandey et al., 2017). Therefore, to test if peroxisome-dependent restriction of Mtb is independent of NADPH oxidase, we targeted gp91-phox (referred to as CYBB), an essential subunit of Nox2. We knocked out CYBB and PEX3 in primary human monocyte-derived macrophages (HMDM) using CRISPR/Cas9 (Fig. 5 A). We nucleofected HMDM with Cas9 protein and single guide RNA (sgRNA) targeting PEX3 and CYBB to obtain a KO pool for the two genes, referred to as PEX3$^{nf}$ and CYBB$^{nf}$ (Fig. 5, B and C). We infected HMDM not nucleofected (referred to as CTRL), CYBB$^{nf}$, and PEX3$^{nf}$ with Mtb WT or Mtb ΔRD1 and then analyzed them by high-content single-cell live imaging. Strikingly, we observed a higher replication rate for both Mtb WT and Mtb ΔRD1 in HMDM lacking CYBB than in CTRL HMDM (Fig. 5 D). Confirming our

observations in iPSDM, the HMDM PEX3$^{nf}$ showed increased Mtb WT but not Mtb ΔRD1 replication after 72 h of infection when compared with CTRL HMDM (Fig. 5 D). Altogether, live cell imaging experiments in both iPSDM and HMDM confirmed that peroxisomal function is implicated in restricting the replication of Mtb WT in a localization-dependent manner distinct from phagosomal NADPH oxidase activity.

Our data show that human macrophages respond to infection with virulent Mtb by inducing peroxisome biogenesis. This increase in peroxisome numbers and volume results in the generation of ROS in peroxisomes and the cytosol to restrict bacteria that escape the phagosome. This mechanism is ineffective in restricting bacteria that primarily reside within phagosomes since the mutant strain, unable to damage the phagosome, is still restricted in PEX3$^{-/-}$ iPSDM. Previous reports have suggested that peroxisome function is required for intracellular bacterial control (Di Cara et al., 2018; Behera et al., 2022) and our study identifies one of the mechanisms by which peroxisomes can contribute to controlling bacterial infection in human macrophages.

It has been reported that peroxisome biogenesis is induced by the Mtb acetyltransferase (Rv3034c) to inhibit ROS in mouse macrophages (Behera et al., 2022). In contrast, we observed that the increase of peroxisomal number is ESX-1 dependent and functionally relates to $H_2O_2$ production in the cytosol (Lismont et al., 2019a), as we accurately monitored $H_2O_2$ with specific genetically encoded probes. These differences in peroxisome modulation and ROS regulation could be attributed to the different in vitro models used. As we employed human cells and a genetic approach to deplete peroxisomes, our findings provide a distinct perspective compared with previous studies. Furthermore, pharmacological inhibition of ROS production would be necessary to validate our findings; however, at this stage, two major bottlenecks preclude these experiments: the lack of spatial control over ROS inhibition and the fact that the ROS inhibitor will affect bacterial growth, affecting the interpretation of the experiments.

It has been proposed that ROS play an important role in the immune response to tuberculosis since NADPH oxidase (Nox) deficiencies, as in chronic granulomatous disease, lead to more severe mycobacterial infections with the attenuated Bacillus Calmette-Guerin and impaired granuloma formation (Deffert et al., 2014b). Nox2 isoform mediates the oxidative burst

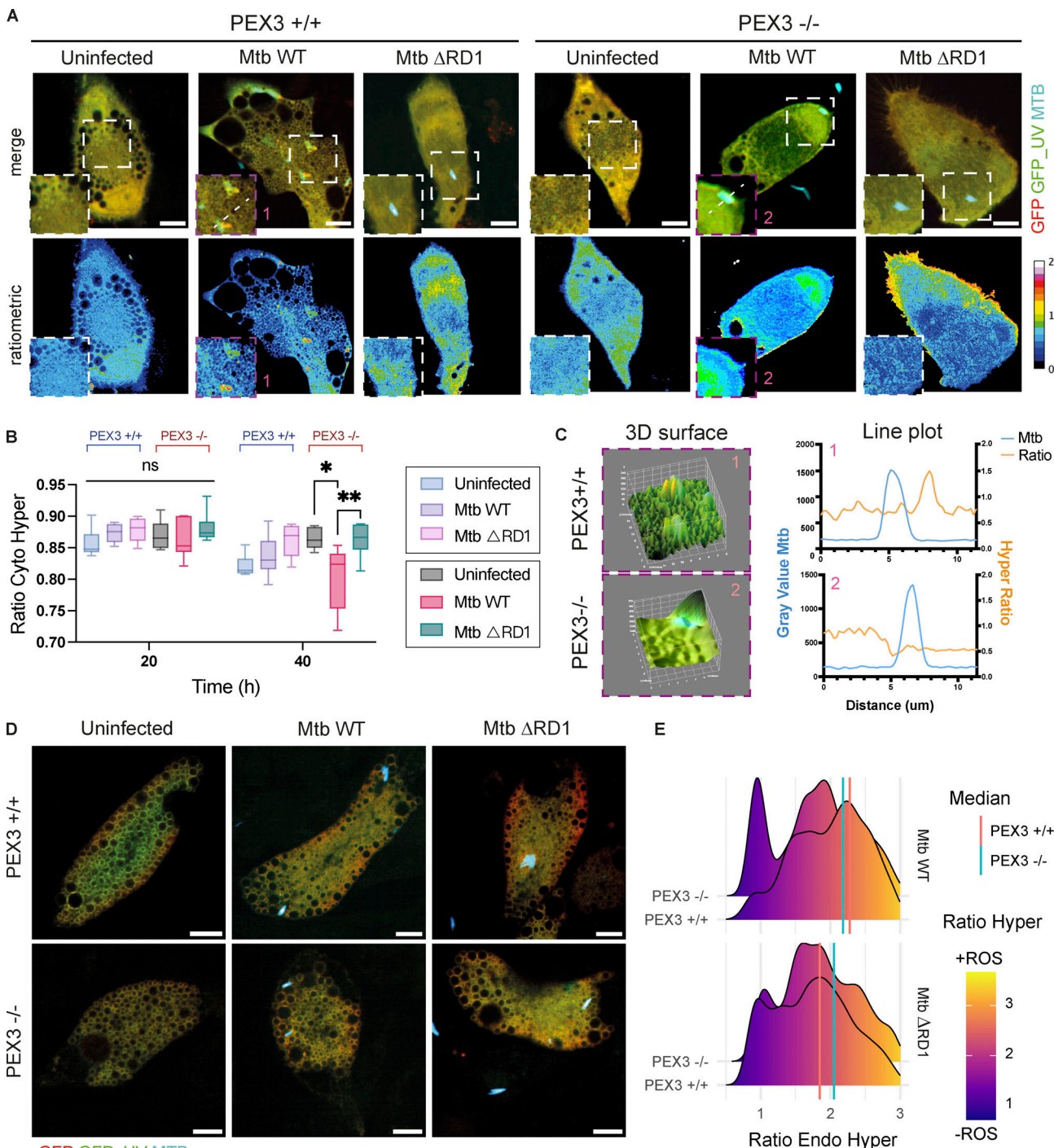

Figure 4. **Peroxisome-dependent restriction of Mtb is associated with higher levels of ROS in the cytosol. (A–C)** Cyto_Hyper reporter during Mtb infection. **(A)** Confocal images of iPSDM at 24 hpi of infection (Uninfected, Mtb WT, and ΔRD1). The left graph shows Cyto_Hyper in iPSDM PEX3[+/+] and right graph shows Cyto_Hyper in PEX3[−/−] iPSDM. Top: Merge of GFP_UV (green), GFP (red), and Mtb-E2-Crimson (cyan). Bottom: Ratiometric imaging of the Cyto_Hyper reporter. Scale bars: 10 μm. **(B)** Quantification of Cyto_Hyper ratio in iPSDM PEX3[+/+] and PEX3[−/−] during infection at 20 and 40 hpi. Data representative from one out of two independent experiments ($n = 4$ independent wells per replicate). Significance was determined using unpaired by two-way ANOVA with Šidák's multiple comparisons post-test. P value (APA) 0.033 (*), 0.002 (**). **(C)** 3D surface (left) and line plot (right) of Cyto_Hyper reporter in iPSDM PEX3[+/+] (1 box) and PEX3[−/−] (2 box) infected with Mtb WT. **(D and E)** Endo_Hyper reporter during Mtb infection. **(D)** Snapshot of Endo_Hyper reporter in iPSDM PEX3[+/+] and PEX3[−/−] during infection over 24 hpi. Merge of GFP_UV (green), GFP (red), and Mtb-E2-Crimson (cyan). Scale bars: 10 μm. **(E)** Analysis of Endo_Hyper ratio around Mtb (area 0.5 μm) in iPSDM PEX3[+/+] and PEX3[−/−] infected with Mtb WT and ΔRD1 at 24 hpi. The red line represents the median of Mtb-Endo_Hyper ratio in PEX3[+/+] and the light blue line represents the median of Mtb-Endo_Hyper ratio in PEX3[−/−] iPSDM. $N > 500$ Mtb regions of interest were quantified per each condition.

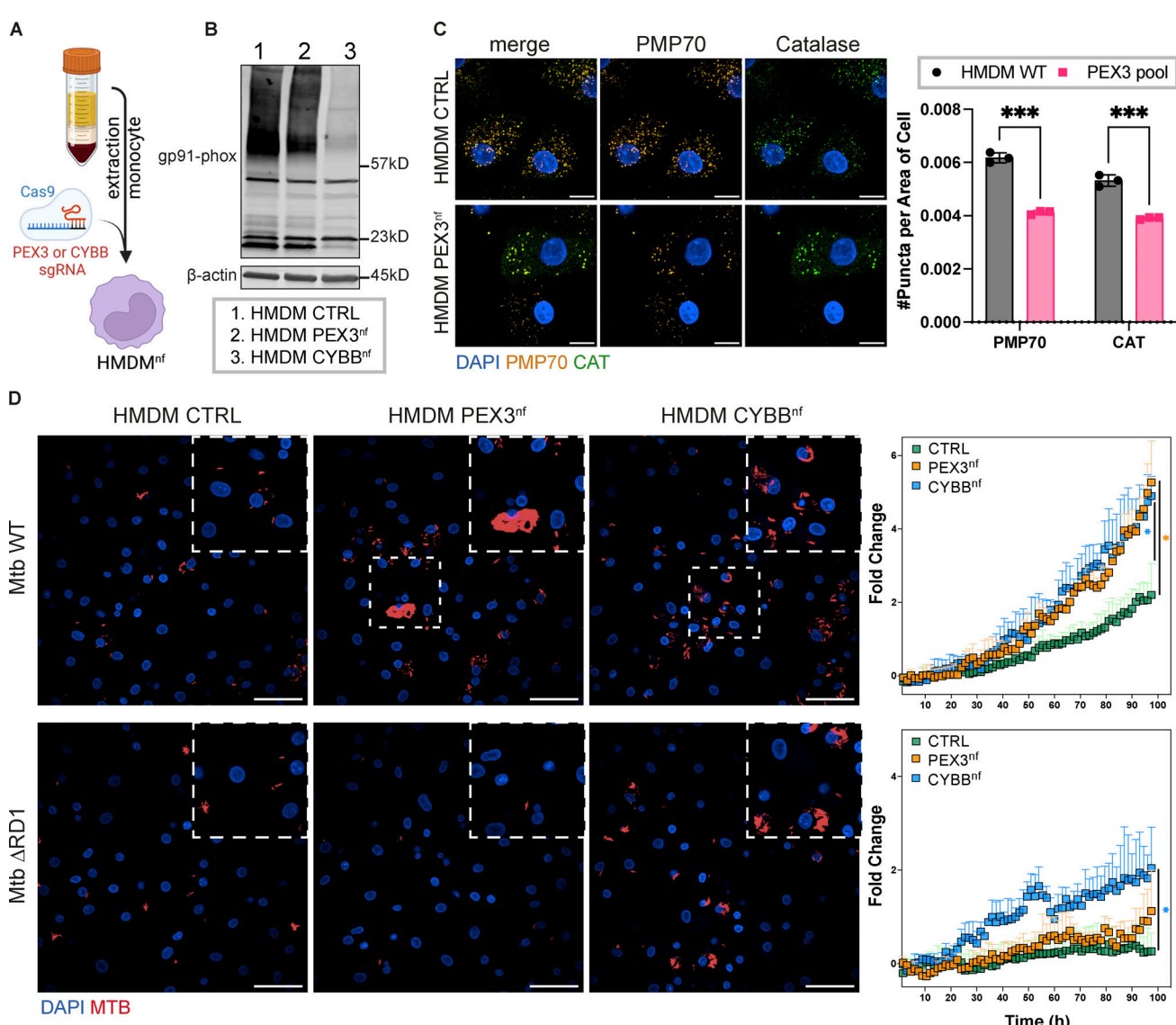

Figure 5. **Peroxisome-dependent restriction of Mtb is spatial-temporally separated from the NADPH oxidase activity in human macrophages.** **(A)** Workflow for nucleofection (nf) approach to KO CYBB (CYBB[nf]) and PEX3 (PEX3[nf]) genes in HMDM. **(B)** Western blot of HMDM CTRL, PEX3[nf], and CYBB[nf] for gp91-phox protein (CYBB). **(C)** Left: Immunofluorescence of HMDM CTRL and PEX3[nf]. Nuclear staining (blue), PMP70 (orange), and CAT (green). Scale bars: 20 µm. Right: Quantification of CAT and PMP70 puncta area per area of cells. Significance was determined by two-way ANOVA with Šidák's test. P value <0.0001 (***). **(D)** Mtb growth in HMDM CTRL, PEX3[nf], and CYBB[nf]. Left: Confocal images of HMDM CTRL, PEX3[nf], and CYBB[nf] at 60 hpi infected with Mtb WT (top) and ΔRD1 (bottom). Nuclear staining (blue) and Mtb-E2-Crimson (red). Scale bars: 100 µm. Right: Fold change of Mtb growth in HMDM CTRL (green), PEX3[nf] (orange), and CYBB[nf] (light blue) over 98 h of infection. The top graph shows the fold change of Mtb WT and the bottom graph the fold change of Mtb ΔRD1. Data representative from one out of two independent experiments (n = 3 independent wells per replicate). Significance was determined only for the last time point (98 hpi) by two-way ANOVA with Dunnett's post-doc test. P value 0.033 (*).

observed after the engulfment of bacterial pathogens and maturation of the resulting phagosome into a phagolysosome in immune cells (Rada and Leto 2008; Soldati and Neyrolles 2012). In contrast, Mtb is able to counteract ROS by expressing factors such as the "enhanced intracellular survival" gene or katG, which have been described to modulate autophagy and inflammatory responses, and suppress host ROS production in macrophages (Dan Dunn et al., 2015; Shin et al., 2010; Mohanty et al., 2015; Ng et al., 2004). Considering that Nox2-derived and mitochondria-derived ROS mostly target phagosomes (Geng et al., 2015; Köster et al., 2017) and this can be suppressed by

Mtb (Köster et al., 2017; Miller et al., 2010), our data uncovers a novel ROS-dependent mechanism that operates in the cytosol that is separate from the NADPH oxidase activity. There are multiple antibacterial mechanisms that recognize and restrict bacteria in the cytosol, and these mechanisms operate by redirecting bacteria to membrane-bound compartments (Bernard et al., 2020) or directly lyse cytosolic bacteria (Gaudet et al., 2021). Our data show that in addition to these membrane trafficking-dependent mechanisms that recognize bacteria in the cytosol, peroxisomes add another layer of control by restricting bacteria in a ROS-dependent manner.

Our data also show that human macrophages lacking the NADPH oxidase are unable to restrict both Mtb WT and Mtb ΔRD1. In contrast, macrophages lacking peroxisomes are more permissive to Mtb WT, but not Mtb ΔRD1, which is localized in phagosomes. Nox2, one isoform of the NADPH oxidase, is crucial for the antimicrobial activity of macrophages (Gluschko et al., 2018; Pollock et al., 1995), but its action is mainly restricted to phagosomes (Köster et al., 2017; Deffert et al., 2014a, 2014b). Therefore, our data show that peroxisomes could represent an additional ROS-dependent mechanism that contributes to mycobacterial control in addition to Nox2, acting as a source of $H_2O_2$ instead of a sink for peroxisomal ROS (Lismont et al., 2019b). Altogether, our work identifies a ROS-dependent antimycobacterial function for peroxisomes in the cytosol of human macrophages, and further research will define if this mechanism applies to other bacteria that access the cytosol of host cells. Targeting pathways that restrict Mtb replication, as the one described here, could be envisioned as part of novel host-directed therapies.

## Materials and methods

### Human iPSC culture and iPSDM differentiation

KOLF2 human iPSC, from a healthy male donor, were sourced from Public Health England Culture Collections (catalog number, 77650100) and maintained in Vitronectin XF (100-0763; StemCell Technologies)–coated plates with Essential E8 medium (A1517001; Gibco). Cells were authenticated by short tandem repeat profiling upon receipt and were checked monthly for mycoplasma contamination by PCR. Cells were passaged 1:6 once at 70% confluency using Versene (15040066; Gibco). Monocyte factories were set up following a previously reported protocol (van Wilgenburg et al., 2013). Briefly, a single-cell suspension of iPSC was produced with TryplE (12604013; Life Technologies) at 37°C for 5 min and resuspended in E8 plus 10 μM Y-27632 (72307; Stem Cell Technologies) and seeded into AggreWell 800 plates (34815; Stem Cell Technologies) with $4 × 10^6$ cells/well and centrifuged at 100 × $g$ for 3 min. The forming embryonic bodies (EBs) were fed daily with two times 50% medium changes with E8 supplemented with 50 ng/ml hBMP4 (120-05; Peprotech), 50 ng/ml hVEGF (100-20; Peprotech), and 20 ng/ml hSCF (300-07; Peprotech) for 3 d. On day 4, the EBs were harvested and seeded at 100–150 EBs per T175 or 250–300 per T225 flask in XVIVO-factory medium (X-VIVO 15, BE02-060F; Lonza; 2 mM GlutaMAX, 35050-038; Gibco; 50 μM β-mercaptoethanol, 21985023; Gibco; 100 ng/ml hM-CSF, 300-25 and 25 ng/ml hIL-3; Peprotech, 200-03; Peprotech). These monocyte factories were fed weekly for 5 wk until plentiful monocytes were observed in the supernatant. Up to 50% of the supernatant was harvested weekly and centrifuged at 300 × $g$ for 5 min. The cells were resuspended in XVIVO-differentiation media (X-VIVO 15; BE02-060F; Lonza; 2 mM GlutaMAX, 35050-038; Gibco; 50 μM β-mercaptoethanol, 21985023 and 100 ng/ml hM-CSF; Gibco, 300-25; Peprotech). Monocytes were plated at $4 × 10^6$ cells per 10-cm Petri dish to differentiate over 7 d, and on day 4, a 50% medium change was performed. To detach cells, iPSDM plates were washed once with PBS (pH 7.4) and then incubated with Versene for 15 min at 37°C and 5% $CO_2$ before diluting 1:3 with PBS and gently scraping. Macrophages were centrifuged at 300 × $g$ and plated for experiments.

### Preparation and culture of HMDMs

Human monocytes were prepared from leucocyte cones (NC24) supplied by the NHS Blood and Transplant service (Lerner et al., 2017). White blood cells were isolated by centrifugation on Ficoll-Paque Premium (17-5442-03; GE Healthcare) for 60 min at 300 × $g$. Mononuclear cells were collected and washed twice with MACS rinsing solution (130-091-222; Miltenyi) to remove platelets and red blood cells. The remaining samples were incubated with 10 ml RBC lysing buffer (R7757; Sigma-Aldrich) per pellet for 10 min at room temperature. Cells were washed with rinsing buffer and were resuspended in 80 μl MACS rinsing solution supplemented with 1% BSA (130-091-376; MACS/BSA; Miltenyi) and 20 μl anti-CD14 magnetic beads (130-050-201; Miltenyi) per $10^8$ cells. After 20 min on ice, cells were washed in MACS/BSA solution and resuspended at a concentration of $10^8$ cells/500 μl in MACS/BSA solution and further passed through an LS column (130-042-401; Miltenyi) in the field of a QuadroMACS separator magnet (130-090-976; Miltenyi). The LS column was washed three times with MACS/BSA solution, then CD14 positive cells were eluted, centrifuged, and resuspended in complete RPMI 1640 with GlutaMAX and Hepes (72400-02; Gibco) and 10% fetal bovine serum (FBS; F7524; Sigma-Aldrich).

### Mycobacterial strains and culture conditions

Mtb H37Rv (Mtb WT) and H37Rv ΔRD1 were kindly provided by Prof. Douglas Young (The Francis Crick Institute, London, UK). Fluorescent Mtb strains were generated as previously reported (Lerner et al., 2016). E2Crimson Mtb was generated by transformation with pTEC19 (30178; Addgene, deposited by Prof. Lalita Ramakrishnan). Strains were verified by sequencing and tested for phthiocerol dimycocerosate positivity by thin-layer chromatography of lipid extracts from Mtb cultures. Mtb strains were cultured in Middlebrook 7H9 (M0178; Sigma-Aldrich) supplemented with 0.2% glycerol (G/0650/17; Fisher Chemical), 0.05% Tween-80 (P1754; Sigma-Aldrich), and 10% ADC (212352; BD Biosciences).

### Macrophage infection with Mtb

The day before infection, iPSDM were seeded at a density of 50,000 cells per well of a 96-well plate; 150,000 cells per well of a 24-well plate; 500,000 cells per well of a 12-well plate; and $1 × 10^6$ cells per well of a 6-well plate. Mid-logarithmic phase bacterial cultures ($OD_{600}$ 0.5-1.0) were centrifuged at 2,000 × $g$ for 5 min and washed twice in PBS. Pellets were then shaken vigorously for 1 min with 2.5–3.5 mm glass beads (332124G; VWR) and bacteria were resuspended in 10 ml macrophage culture media before being centrifuged at 300 × $g$ for 5 min to remove large clumps. The top 7 ml of bacterial suspension was taken, $OD_{600}$ recorded, and diluted appropriately for infection. The inoculum was added at the correct MOI, assuming $OD_{600}$ of 1 is $1 × 10^8$ bacteria/ml. Infections were carried out in a volume of 50 μl in a 96-well plate, 300 μl in a 24-well plate, or 500 μl in a 12-well plate. After 2 h of uptake, extracellular bacteria were

removed with two washes in PBS and macrophages were incubated at 37°C and 5% $CO_2$ for the required time points in macrophage media. At the required time after infection, cells were harvested or fixed in 4% PFA as appropriate. An MOI of 1 was used for replication experiments. For all other experiments, cells were infected with an MOI of 2.

## Plasmids
All DNA constructs were produced using *Escherichia coli* DH5a (Thermo Fisher Scientific) and extracted using a plasmid midiprep kit from Qiagen. The plasmids used in this study were mEGFP-N1 (plasmid #54767; Addgene); pHyPer-cyto (FP941; Evrogen); and pHyPer-pexo, pHyPer-endo, PEX3_Turbo, and pAAVS_Nst_CAG_mEGFP_PTS1 generated in our lab.

## Cloning of pHyPer-pexo and pHyPer-endo vector
To generate the pHyPer-pexo (referred to as Pexo_Hyper), pHyPer-cyto (FP941; Evrogen) was used as a template. The primer Fw_Hyper_BamHI and Rv_Hyper_PTS1_NotI (Table S1) were used to perform PCR. The PCR product was extracted using the clonetech nucleospin gel and PCR cleanup kit (Cat. #740609.5; Takara Clonetech) according to the vendor's instructions. Both the PCR product and the mEGFP-N1 (Plasmid #54767; Addgene) were digested with BamHI and NotI restriction enzyme and the resulting product was ligated using the DNA Ligation Kit, Mighty Mix (Cat. #6023; Takara Clonetech).

To generate pHyPer-endo (referred to as Endo_Hyper), +HyPer7 (plasmid #136466; Addgene) and the Lamp1-mGFP (plasmid #34831; Addgene) were digested with BamhI and XbaI restriction enzyme. The Hyper7 sequence was then cloned into the Lamp1 backbone by using the DNA Ligation Kit.

## Cloning of PEX3_Turbo vector
For the PEX3_Turbo, the PEX3 sequence was amplified by PCR with the PEX3_cDNA_F_infusion and PEX3_cDNA_R_TURBO_infusion as primers (Table S1) and the PEX3 cDNA ORF (HG14106-UT; SinoBiological) as template. The In-Fusion cloning reaction was performed using the purified PCR product and the tdTurboRFP-Lysosomes-20 (plasmid #58061; Addgene) as backbone, following the vendor's instructions (In-Fusion HD Cloning Plus, #638910).

## Cloning of pAAVS_Nst_CAG_mEGFP_PTS1 vector
To generate the pAAVS_Nst_CAG_mEGFP_PTS1 vector, the pENTR_mEGP_PTS1 was generated at first. Initially, the mEGFP-N1 (plasmid #54767; Addgene) was digested by BsrgI/NotI, and a double oligo (top PST1 and bottom PST1, see Table S1) with overhang was used for sticky-end ligation. The new plasmid, called mEGFP-N1-PTS1, was then digested with BamHI and XbaI and the fragment containing the mEGFP_PTS1 sequence was cloned into the pENTR2B_kan_adaptor (gift from Prof. William Skarnes from The Jackson Laboratory, Farmington, CT, USA), previously digested with BamHI and XbaI. Once the sequence of the pENTR_mEGP_PTS1 was confirmed by Sanger sequencing, the plasmid was recombined using LR clonase (Gateway LR Clonase II Enzyme mix, Cat. #11791020; Thermo Fisher Scientific) into the destination vector pAAVS1_Nst_CAG_DEST (plasmid #80489; Addgene).

## mEGFP_PTS1 knock-in in the AAVS1 locus of human iPSC
The mEGFP_PTS1 gene, under the control of a strong constitutive promoter (CAG), was inserted by homologous recombination in one copy of the AAVS1 locus of KOLF2 cells using a highly efficient AAVS1 targeting system (Oceguera-Yanez et al., 2016). Co-delivery of 20 µg of Cas9, 16 µg of sgRNA (AAVS1_sg2), and 2 µg of circular, supercoiled plasmid (pAAVS_Nst_CAG_mEGFP_PTS1) was performed by nucleofection, and G418-resistant colonies were assayed for correct targeting of CAG-mEGFP_PTS1 into AAVS by immunofluorescence.

## PEX3 gene KO in human iPSC
CRISPR/Cas9 technology was used to generate PEX3 KO iPSCs. The KO strategy was based on using four sgRNAs flanking specific gene exons to obtain deletion of a genomic sequence. The sgRNAs targeting *PEX3* were designed and selected considering the lowest off-target score by using WGE CRISPR design tool (http://www.sanger.ac.uk/htgt/wge/; Hodgkins et al., 2015). Nucleofection of KOLF2 iPSCs was performed by using the Amaxa 4D-Nucleofector (V4XP-3024; Lonza). For each nucleofection, $1 \times 10^6$ of human iPSCs were resuspended in 100 µl of P3 buffer (V4XP-3024; Lonza) containing 20 µg of S.p. Cas9 (Alt-R S.p. Cas9 Nuclease V3, 1081059, IDT) mixed with a total of 16 µg of targeting synthetic chemically modified sgRNAs (Synthego; Table S2). The cells and the Cas9-RNP mix were then nucleofected with the CA-137 program. After nucleofection, single clones were manually picked (Skarnes et al., 2019) and screened by PCR-based assay (see Table S3 for sequences). Selected PCR-validated KO clones were expanded, and immunofluorescence was performed to assess the loss of the corresponding protein and the loss of peroxisomal structure.

## iPSDM electroporation
Plasmid DNA was electroporated into iPSDM using the Neon system (Invitrogen). iPSDM were resuspended at $1.5 \times 10^6$ cells in 100 µl buffer R. 10 µl of cell/1 µg plasmid DNA mix was aspirated into a Neon pipette and electroporated in electroporation buffer "E" at 1,500 V for 30 ms with one pulse. Cells were then plated for imaging studies.

## Flow cytometry
Cells were collected and incubated in PBS plus 0.1% BSA (9998S; Cell Signalling Technologies) and 5 µl Fc block per million cells for 20 min. 50 µl of cells were then incubated with 50 µl antibody cocktail diluted in PBS and 0.1% BSA for 20 min on ice in the dark. Cells were washed in 2 ml PBS and fixed in 2% PFA (15710; Electron Microscopy Sciences) diluted in PBS prior to analysis. Cells were analyzed on an LSRII flow cytometer. Antibodies were purchased from BD Biosciences Antibody (CD14-Alexa488, 562689; CD119-PE, 558934; CD86-BV421, 562433; CD11b-bv421, 562632; CD163-FITC, 563697; CD169-PE, 565248; CD206-APC, 561763; CD16-Alexa647, 557710; Alexa488 isotype, 557703; Alexa647 isotype, 57714; PE isotype, 12-4015-82; BV421 isotype, 562438; CD16-Alexa647, 557710; Alexa488 isotype, 557703). Flow cytometry data was analyzed and plotted in FlowJo (BD Biosciences).

## Nucleofection of HMDM

Human monocytes were washed twice with PBS and electroporated in the appropriate primary nucleofection solution (Cat. No. VPA-1007; Amaxa Human Monocyte Nucleofector Kit) using the Lonza 2b Nucleofector (AAB-1001; Nucleofector 2b Device). $5 \times 10^6$ of human monocytes were used per reaction and resuspended in 100 µl of primary nucleofection solution containing 4 µg of S.p. Cas9 (IDT) mixed with a total of 12 µg of targeting synthetic chemically modified sgRNAs (Synthego; Table S2). Human monocytes were then nucleofected with the sgRNA pool and the Cas9-RNP mix using the Y001 program. Nucleofected cells were cultured in prewarmed RPMI 1640 supplemented with GlutaMAX, Hepes, and 10% FBS in a 6-well plate. 2 h after nucleofection, 100 ng/ml hM-CSF was added to the cells. Dishes were incubated in a humidified 37°C incubator with 5% $CO_2$. After 3 d, an equal volume of fresh complete media including 100 ng/ml hM-CSF was added. 6 d after the initial isolation, differentiated macrophages were detached in 0.5 mM EDTA in ice-cold PBS using cell scrapers (83.1830; Sarsted), pelleted by centrifugation, and resuspended in RPMI medium containing 10% FBS (Hiatt et al., 2021). Cells were seeded at a density of 60,000 cells per well of a 96-well plate and 600,000 cells per well of a 12-well plate.

## SDS-PAGE and Western blot

Cells were lysed in radioimmunoprecipitation assay buffer (20-188; Millipore) containing protease and phosphatase inhibitor cocktail (78445; Thermo Fisher Scientific) for 10 min on ice. LDS sample buffer (NP008; Thermo Fisher Scientific) and NuPage Sample Reducing Agent (NP009; Thermo Fisher Scientific) were added and samples were boiled at 95°C for 20 min if infected with Mtb, otherwise 10 min. Samples were loaded into 4–12% Bis-Tris SDS-PAGE gels (WG1403BOX, NP0322BOX, NP0321BOX; Thermo Fisher Scientific), and electrophoresis was carried at 100 V for 120 min. Proteins were transferred to polyvinylidene difluoride membranes (IB24002, IB24001; Thermo Fisher Scientific) on an iBlot2 (IB21001; Thermo Fisher Scientific) using program P0. Membranes were blocked in 5% skimmed milk (B008KK2DMK; VWR) in TBS + 0.01% Tween 20 (TBS-T) for 1 h at room temperature with shaking. Primary antibodies (1:1,000), diluted in 5% BSA in TBS-T, were incubated with membranes overnight at 4°C with shaking. Blots were washed three times in TBS-T and incubated with HRP-conjugated secondary antibodies (1:5,000) in 5% skimmed milk in TBS-T for 1 h at room temperature. Blots were developed with ECL (WBULF0500; Millipore) and imaged on a GE Amersham Imager 680 (GE Healthcare). The molecular weight ladder (116028; Abcam) was used for determining the approximate size of a protein. Antibodies used were PEX14 (ab183885; Abcam), PMP70 (ab211533; Abcam), ACOX1 (ab184032; Abcam), HSD17B4 (ab128565; Abcam), Catalase (ab16731; Abcam), ABCD1 antibody (ab197013; Abcam), PEX11B (ab181066; Abcam), Pex19 (PA5-22129; Thermo Fisher Scientific), ACSL1 (#9189; Cell Signaling), gp91-phox (SC-130543; Santa Cruz Biotechnology), PEX3 (10946-1-AP; Proteintech), β-actin-HRP (12262) from Cell Signaling Technologies, anti-rabbit-IgG conjugated to HRP (W4011), and anti-mouse-IgG conjugated to HRP (W4021) from Promega.

## Indirect immunofluorescence

Cells were fixed in 4% PFA diluted in PBS and kept overnight at 4°C. The samples were quenched with 50 mM $NH_4Cl$ in PBS for 10 min at room temperature and permeabilized with 0.3% Triton X-100 and 5% FBS in PBS for 30 min. Antibodies were diluted in PBS containing 5% FBS and incubated for 1 h at room temperature. Between antibodies, cells were washed three times in PBS. Nuclei were stained for 10 min with DAPI (D1306; Thermo Fisher Scientific) diluted 1:10,000 in PBS. Coverslips were mounted on glass slides with DAKO mounting medium (DAKO, S3023). Antibodies used were CAT (D4P7B) XP (12980, 1:700; Cell Signaling), PEX14 (ab183885, 1:400; Abcam), PMP70 (ab211533, 1:400; Abcam), Tom20 (#11802-1-AP, 1:100; Invitrogen), OCT3/4 (SC5279, 1:100; Santa Cruz), TRA-1-60 (#MAB4360, 1:100; Millipore), TRA-1-81 (#MAB4381. 1:100; Millipore), anti-rabbit-Alexa Fluor 488 (A11034, 1:500; Life Technologies), Donkey anti-mouse IgG-594 (A21203, 1:300; Thermo Fisher Scientific), and anti-mouse Alexa Fluor 546 (A11003, 1:500; Life Technologies).

## Confocal microscopy

Coverslips were imaged using a Leica SP8 inverted confocal microscope (Leica Microsystems) with a 63× 1.4NA oil immersion objective. For imaging on the Leica SP8, DAPI was excited at 405 nm, Alexa Flour 488 or mEGFP was excited at 488 nm, Alexa Flour 546 or Turbo was excited at 561 nm, and E2-Crimson was excited at 633 nm. Fluorescence was detected using HyD detectors. Laser and detector settings were kept constant between conditions for each biological replicate of an experiment.

## High-content live-cell imaging

High-content live-cell imaging: 50,000 iPSDM were seeded into a 96-well glass bottom Viewplate (6005430; Perkin Elmer) or olefin-bottomed 96-well plate (6055302; Perkin Elmer) and infected with Mtb as described above. The plate was sealed with parafilm and placed in a preheated (37°C) Opera Phenix microscope with either a 40× 1.1NA or 63× 1.15NA water immersion objective (Perkin Elmer) with 5% $CO_2$. Images were acquired in confocal mode, a binning of 1. Capture settings were: Mtb E2crimson was excited with the 640 nm laser at 10% power with 100 ms exposure. The pHyPer-cyto, pHyPer-pexo, and pHyPer-endo construct were excited with the 405 and 488 nm lasers, and emission was collected at 510 nm for both excitations. At least 20 fields per well were imaged in all the experiments. Images were acquired at 1,020 × 1,020 pixels using Harmony 4.9 high content imaging and analysis software (PerkinElmer). Hoechst H33342 (H3570; Thermo Fisher Scientific) was excited using the 405 nm laser at 15% power with 100 ms exposure. Fluorescence was detected using a 16-bit scMOS camera.

## Pharmacological treatment

iPSDM were treated, the time is described in the text, with 4 mM of 4-PBA (#2682; BIO-TECHNE), 40 mM of 3-AT (#A8056; Sigma-Aldrich), and 15 µM of GW (#1677; Tocris).

## RNA isolation and quantitative PCR (qPCR)

Total RNA was prepared from iPSDM cells with the RNeasy Mini Kit (#74104; Qiagen) and DNA was removed by on-column

digestion with rDNase. cDNA was synthesized from 2 µg RNA with random hexamer primers using the QuantiTect Rev. Transcription Kit (#205313; Qiagen) and diluted 10-fold in sterile H20. The real-time qPCR reaction was performed on Applied Biosystems QuantStudio 7 Flex Real-Time PCR System in triplicate using 10 ng cDNA, 7.5 pmol forward and reverse primers, and the 2× TaqMan Universal PCR Master Mix Thermo Fisher Scientific (#4304437; Life Technologies). The relative mRNA amount was calculated using the comparative threshold cycle method. GADPH was used as the invariant control. For details of qPCR primers used, see Table S4, as shown by Michael Schrader's group (Azadi et al., 2020).

### High-content imaging of Mtb replication
After 2, 24, 48, or 72 h of infection, cells were stained for 15 min using DAPI (D1306; Thermo Fisher Scientific) diluted 1:10,000 in PBS. Cell imaging was performed using the OPERA Phenix microscope with 40× 1.1NA water-immersion objective with a 10% overlap between adjacent fields, three wells per condition per experiment. For imaging on the OPERA Phenix, DAPI was detected using λex = 405 nm/λem = 435–480 nm and E2-Crimson bacteria was detected using λex = 640 nm/λem = 650–760 nm. Segmentation and analysis were performed using the Harmony software (version 4.9; Perkin Elmer) where maximum projection of individual z-planes with an approximate distance of 1 µm was used to perform single-cell segmentation by combining the "Find nuclei" and "Find cells" building blocks. For quantifying Mtb replication, bacteria were detected by the "Find spots" building block of Harmony. To determine the bacteria area for each cell, the spot area was summed for each segmented cell. The mean bacteria area per cell of each time point and condition were analyzed by R Studio Software (The R Project for Statistical Computing, version 1.3.1073). Mtb growth as fold change (growth index) was calculated by the formula: (mean Mtb area per cell at t [timepoint of interest] h – mean Mtb area per cell at t2 h)/(mean Mtb area per cell t2 h).

### Long-term live-cell imaging of Mtb replication and HMDM
For live-cell imaging, 55,000 macrophages were seeded per well on an olefin-bottomed 96-well plate (6055302; Perkin Elmer). Cells were infected with Mtb at an MOI of 1 for 2 h. After infection, cells were washed with PBS and replaced with a macrophage media. Imaging was performed using the OPERA Phenix microscope with 40× 1.1 NA water-immersion objective with a 10% overlap between adjacent fields. Five planes with 1 µm distance of more than 20 fields of view were monitored in time and snapshots were taken every 1.5 h for 96 h. For imaging on the Opera Phenix, Brightfield was detected using λex = transmission/λem = 650–760 nm, and E2-Crimson bacteria was detected using λex = 640 nm/λem = 650–760 nm using a 16-bit scMOS camera. For assessing bacterial replication, analyses were performed with Harmony software where maximum projection of individual z-planes with an approximate distance of 1 µm was used. To perform cellular segmentation "Find texture regions," building blocks were trained in Brightfield channel to segment cellular areas. Following the segmentation of cellular area Find spots, building blocks were used to segment

Mtb. To determine the bacteria area over time, the spot area was summed for each time point. Mtb growth as fold change was calculated by the formula: (sum of intracellular Mtb area for the time point – sum of intracellular Mtb area t0 h)/(sum of intracellular Mtb area t0 h).

### 3D imaging analysis
Imaris file converter 9.9.1 was used to open and convert the .lif file into .ims. Peroxisomes in a cell were counted with the Imaris 9.9.1 software. Image z stacks were used to create a "surface" for the GFP_PTS1 and PEX14 staining. Each surface was created with the "add new surface" function. Only the region of interest was selected to continue the analysis. The source channel was selected: green for GFP_PTS1 and yellow for PEX14, and a background subtraction was applied ("diameter of larger sphere which fits into the object" = 0.5 µm). The split touching objects were enabled and the "seed point diameter" was set to 0.250 µm and "quality" = 10.

Both surfaces (GFP_PTS1 = Surface 1 and PEX14 = Surface 2) were classified based on "surface distance to surface" with a minimum distance of 0.5 µm, and data were extracted for co-localization analysis.

### Morphology analysis
To analyze peroxisome morphology, the ImageJ (National Institutes of Health) analysis tool 3D Objects Counter was used to threshold and measure surface area and volume of each peroxisome, with a minimum size requirement of 10 consecutive voxels (at a resolution of 0.065 mm/pixel). Volume was shown as micrometer^3.

### Pexo, Cyto, and Endo_Hyper image analysis
The Pexo, Cyto, and Endo_Hyper construct was excited with the 405 and 488 nm lasers and emission was collected at 510 nm for both excitations. At least 20 fields per well were imaged in all the experiments. Images were acquired at 1,020 × 1,020 pixels using Harmony 4.9 high-content imaging and analysis software (PerkinElmer). For the HyPer construct evaluation, the GFP405 exc/GFP488 exc (GFP/GFP_UV) intensity ratios were determined and the mean per cell was quantified. The HyPer transfected cells were single-cell segmented using a Gaussian filter and cell mask building block based on the staining due to incompatibility of the HyPer construct with blue fluorescent nuclear dyes. For the Pexo_Hyper, the peroxisomes were segmented using a find spot building block, and the intensity was calculated as above (GFP/GFP_UV) on only the peroxisome "spots." For the Cyto_Hyper, the Mtb was segmented using a find spot building block and the area was extended for 1 µm. Intensity was calculated as above (GFP/GFP_UV) on only the "extended spots area." For the Endo_Hyper, the Mtb was segmented using a find spot building block and the area extended for 0.5 µm by zone (1–7). Intensity was calculated as above (GFP/GFP_UV) on only the extended spots area.

### Quantification and statistical analysis
Analysis and model fitting was conducted using Prism (Graphpad). Unless otherwise indicated, all data obtained from at least

three independent experiments ($n$ = 3; biological replicates) were combined for statistical analyses. We considered the population number as $N$, representing the total number of cells analyzed. The statistical tests used to analyze the results are indicated in the figure legends. The P values are indicated by the American Psychological Association (APA) style, 0.033 (*), 0.002 (**), <0.0001 (***). P <0.033 was considered to be significant.

## Online supplemental material
This manuscript is accompanied by three supplementary figures. Fig. S1 contains data supporting Fig. 2. It shows the generation and characterization of PEX3$^{-/-}$ iPSCs and iPSDM. Fig. S2 contains data supporting Fig. 2. It shows the rescue and characterization of PEX3$^{-/-}$ iPSDM. In addition, it shows Mtb area during infection in PEX3$^{+/+}$ and PEX3$^{-/-}$ iPSDM. Fig. S3 contains data supporting Figs. 3 and 4. It shows the characterization of the Hyper reporter, the in vitro Mtb growth in the presence of peroxisomal drug modulator, and the RT-PCR of iPSDM treated with 3-AT and GW. Table S1 lists all the cloning primers and oligos sequences used in this study. Table S2 lists all the guide RNAs and their sequences used in this study. Table S3 lists all the PCR and sequencing primers used in this study. Table S4 lists all the qPCR primers used in this study.

## Data availability
The data reported in this article are available in the published article and its online supplemental material. The plasmids generated in this study are available from the corresponding author upon request.

## Acknowledgments
We are grateful to the Human Embryonic Stem Cell Unit and Advanced Light Microscopy for their support in various aspects of the work. We thank Elena Marcassa (The Francis Crick Institute) for help with the Western blot.

This work was supported by the Francis Crick Institute (to M.G. Gutierrez), which receives its core funding from Cancer Research UK (FC001092), the UK Medical Research Council (FC001092), and the Wellcome Trust (FC001092). This project has received funding from the European Research Council under the European Union's Horizon 2020 research and innovation program (grant agreement no. 772022). P. Santucci has received funding from the European Union's H2020 research and innovation program under the Marie Sklodowska-Curie grant agreement SpaTime_AnTB no. 892859 and has been the recipient of a non-stipendiary Federation of European Biochemical Societies long-term fellowship.

Author contributions: M.G. Gutierrez and E. Pellegrino conceived the project. E. Pellegrino performed most of the experiments and analyzed the data. N. Athanasiadi, C. Bussi, B. Aylan, E.M. Bernard, and P. Santucci contributed to the stem cell culture and macrophage production. C. Bussi contributed with the flow cytometry, B. Aylan contributed with the Western Blot, L. Botella helped with the Mtb growth curves, and A. Fearns contributed with imaging. E. Pellegrino and M.G. Gutierrez wrote the initial draft and E. Pellegrino prepared the figures. M.G. Gutierrez and E. Pellegrino revised the manuscript with input from all authors.

Disclosures: The authors declare no competing interests exist.

Submitted: 15 March 2023

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

## Supplemental material

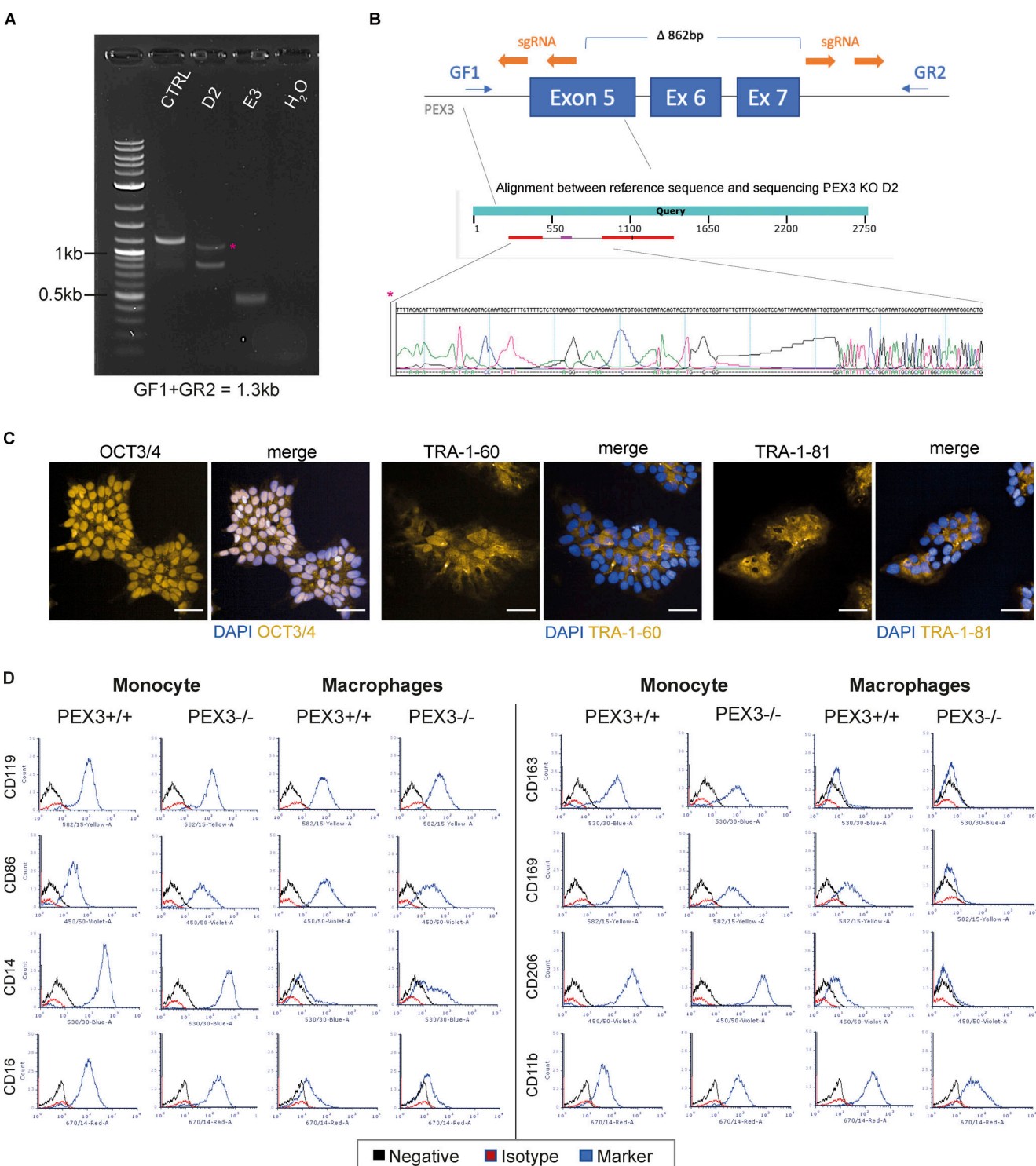

Figure S1. **Generation and characterization of PEX3$^{-/-}$ iPSC and iPSDM clones. (A and B)** Selection of PEX3$^{-/-}$ clones. **(A)** PCR genotyping of the expanded clones, D2 and E3. **(B)** Top: Schematic representation of the PEX3 KO CRISPR strategy. PCR primers (blue), sgRNA (orange). Bottom: Sanger sequencing of the upper band for the D2 clones (*). **(C)** Immunofluorescence of iPSC PEX3$^{-/-}$ for pluripotent markers (OCT3/4, TRA-1-60, and TRA-1-81). Scale bars: 100 µm. **(D)** Flow cytometry characterization of PEX3$^{+/+}$ and PEX3$^{-/-}$ monocytes and macrophages. Names of the markers are indicated on the graph graphs. Black, negative sample; red, isotype control; blue, marker.

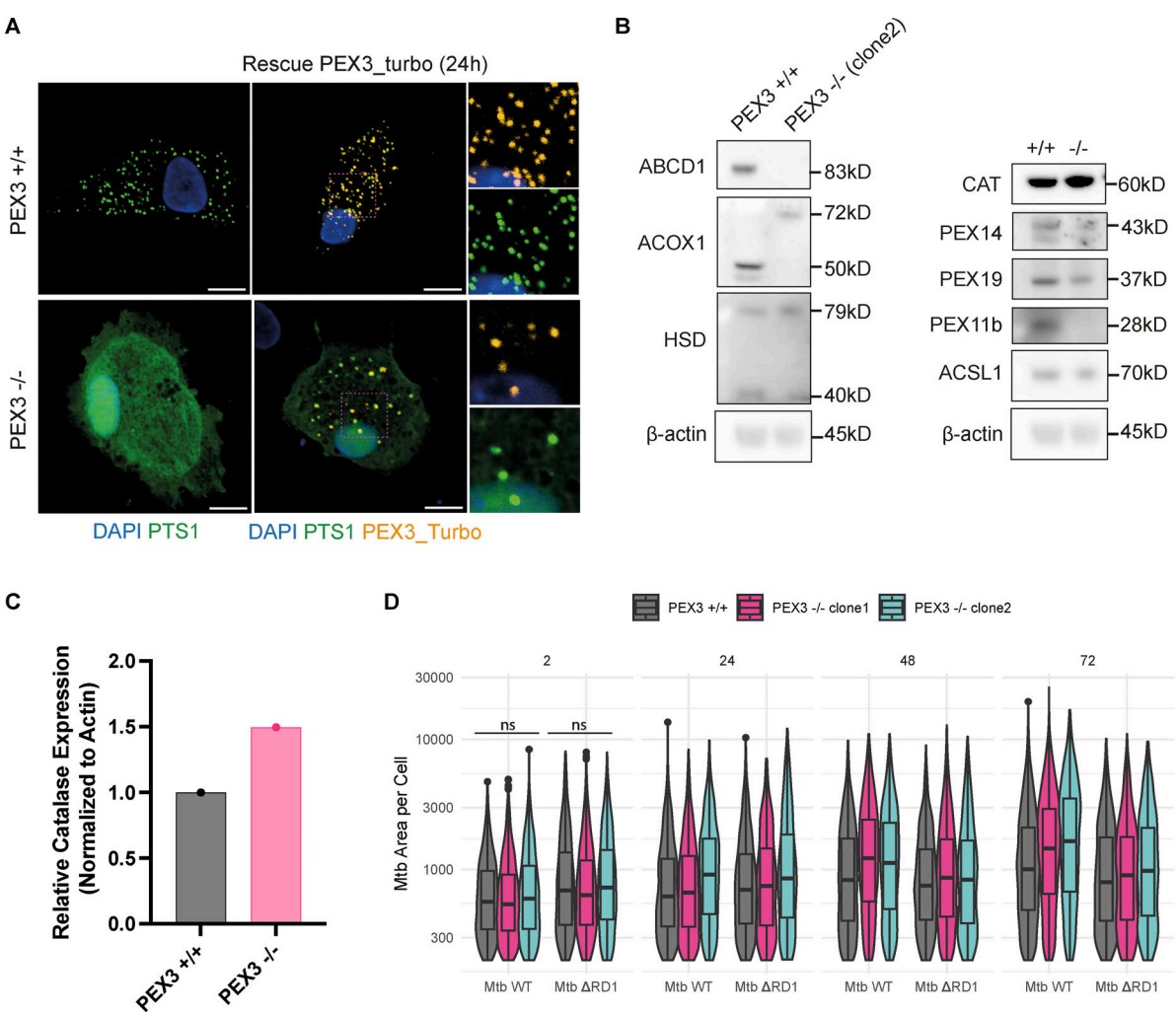

**Figure S2. Characterization of PEX3⁻/⁻ iPSDM clones and uptake and growth of Mtb in PEX3⁺/⁺ and PEX3⁻/⁻ iPSDM. (A)** Rescue experiment with PEX3_turbo for 24 h. Snapshot of iPSDM PEX3⁺/⁺ and PEX3⁻/⁻. Nuclear staining (blue), PEX3_turbo (orange), and EGFP-PTS1 (green). Scale bars: 10 μm. **(B)** Western blot of peroxisomal related protein expressed in iPSDM PEX3⁺/⁺ and PEX3⁻/⁻ at the steady state. **(C)** Quantification of CAT expression from B normalized with actin. **(D)** Analysis of Mtb growth in iPSDM PEX3⁺/⁺ and PEX3⁻/⁻. Violin plot representation of Mtb area (px2) per cells over time (2, 24, 48, and 72 hpi) in PEX3⁺/⁺ and PEX3⁻/⁻ (clone 1 and 2) during infection with Mtb WT and Mtb ΔRD1. Significance was determined for the 2 hpi time point by two-way ANOVA with Tukey's multiple comparison post-test. Source data are available for this figure: SourceData FS2.

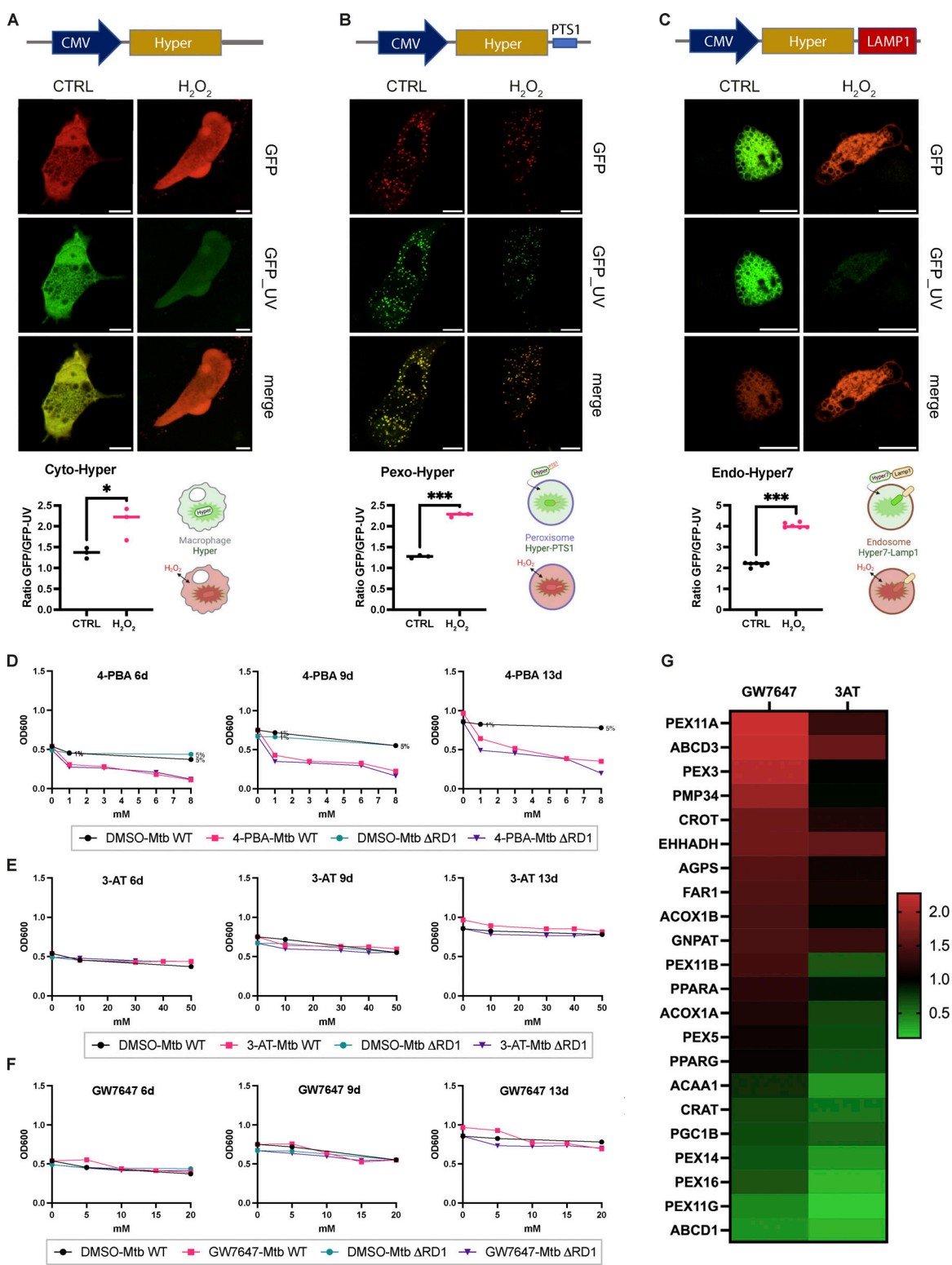

Figure S3. **HyPer reporter: Monitoring peroxisomal (Pexo_Hyper), cytosolic (Cyto_Hyper), and endosomal (Endo_Hyper) H₂O₂.** Related to Figs. 2 and 3. Pharmacological modulation of peroxisomes: minimal inhibitory concentration (MIC) and in vitro characterization of the drugs. **(A)** Schematic representation of the Cyto_Hyper reporter (top) and live snapshot of iPSDM expressing the reporter and treated with H₂O₂ as a positive control (middle). Quantification of the Cyto_Hyper ratio with and without H₂O₂ (bottom). **(B)** Schematic representation of the Pexo_Hyper reporter (top) and live snapshot of iPSDM expressing the reporter and treated with H₂O₂ as a positive control (middle). Quantification of the Pexo_Hyper ratio with and without H₂O₂ (bottom). **(C)** Schematic representation of the Endo_Hyper reporter (top) and live snapshot of iPSDM expressing the reporter and treated with H₂O₂ as a positive control (middle). Quantification of the Endo_Hyper ratio with and without H₂O₂ (bottom). Significance was determined by unpaired *t* test. P value (APA) 0.033 (*), <0.0001 (***). Scale bars: 10 µm. **(D–F)** MIC of 4-PBA (D), 3-AT (E), and GW (F) for Mtb WT and ΔRD1. Mtb growth measured as OD₆₀₀ at 6, 9, and 13 d. **(G)** Heatmap of differentially expressed genes from qPCR analysis of iPSDM treated with either GW or 3-AT for 48 h normalized to the untreated control.

Provided online are four tables. Table S1 shows cloning primers and oligos sequences used in this study. Table S2 shows guide RNAs and their sequences used in this study. Table S3 shows primers and primer sequences used in this study. Table S4 shows human qPCR primers used in this study.

