## [Peer Review File · The Journal of Cell Biology]

Peroxisomal ROS control cytosolic Mycobacterium tuberculosis replication in human macrophages

Enrica Pellegrino, Beren Aylan, Claudio Bussi, Antony Fearn, Elliott Bernard, Natalia Athanasiadi, Pierre Santucci, Laure Botella, and Maximiliano Gutierrez

Corresponding Author(s): Maximiliano Gutierrez, The Francis Crick Institute

Review Timeline:	Submission Date:	2023-03-15
	Editorial Decision:	2023-04-24
	Revision Received:	2023-07-27
	Editorial Decision:	2023-08-23
	Revision Received:	2023-09-05

Monitoring Editor: Craig Roy

Scientific Editor: Andrea Marat

Transaction Report:

DOI: <https://doi.org/10.1083/jcb.202303066>

April 24, 2023

Re: JCB manuscript #202303066

Dr. Maximiliano G Gutierrez
The Francis Crick Institute
1 Midland Road
London NW1 1AT
United Kingdom

Dear Dr. Gutierrez,

Thank you for submitting your manuscript entitled "Peroxisomal ROS control cytosolic Mycobacterium tuberculosis replication in human macrophages". The manuscript was assessed by expert reviewers, whose comments are appended to this letter. We invite you to submit a revision if you can address the reviewers' key concerns, as outlined here.

You will see that the reviewers appreciate the potential significance of your findings and the range of techniques employed. However, they have provided constructive comments which we hope you will agree will further improve your manuscript. In particular, we agree that a revised study must include an analysis of peroxisome abundance in addition to number. Additional details on the statistical analysis and evidence for significance are required throughout. While an analysis on mitochondria would be welcome, this is not required. Otherwise, we expect you to address all remaining reviewer concerns in your revised manuscript.

GENERAL GUIDELINES:

Text limits: Character count for a Report is < 20,000, not including spaces. Count includes title page, abstract, introduction, the joint Results & Discussion, and acknowledgments. Count does not include materials and methods, figure legends, references, tables, or supplemental legends.

Figures: Reports may have up to 5 main text figures. To avoid delays in production, figures must be prepared according to the policies outlined in our Instructions to Authors, under Data Presentation, <https://jcb.rupress.org/site/misc/ifora.xhtml>. All figures in accepted manuscripts will be screened prior to publication.

Supplemental information: There are strict limits on the allowable amount of supplemental data. Reports may have up to 3 supplemental figures. Up to 10 supplemental videos or flash animations are allowed. A summary of all supplemental material should appear at the end of the Materials and methods section.

Please note that JCB now requires authors to submit Source Data used to generate figures containing gels and Western blots with all revised manuscripts. This Source Data consists of fully uncropped and unprocessed images for each gel/blot displayed in the main and supplemental figures. Since your paper includes cropped gel and/or blot images, please be sure to provide one Source Data file for each figure that contains gels and/or blots along with your revised manuscript files. File names for Source Data figures should be alphanumeric without any spaces or special characters (i.e., SourceDataF#, where F# refers to the associated main figure number or SourceDataFS# for those associated with Supplementary figures). The lanes of the gels/blots should be labeled as they are in the associated figure, the place where cropping was applied should be marked (with a box), and molecular weight/size standards should be labeled wherever possible. Source Data files will be made available to reviewers during evaluation of revised manuscripts and, if your paper is eventually published in JCB, the files will be directly linked to specific figures in the published article.

The typical timeframe for revisions is three to four months. While most universities and institutes have reopened labs and allowed researchers to begin working at nearly pre-pandemic levels, we at JCB realize that the lingering effects of the COVID-19 pandemic may still be impacting some aspects of your work, including the acquisition of equipment and reagents. Therefore,

if you anticipate any difficulties in meeting this aforementioned revision time limit, please contact us and we can work with you to find an appropriate time frame for resubmission. Please note that papers are generally considered through only one revision cycle, so any revised manuscript will likely be either accepted or rejected.

Thank you for this interesting contribution to Journal of Cell Biology. You can contact us at the journal office with any questions, cellbio@rockefeller.edu or call (212) 327-8588.

Sincerely,

Craig Roy, PhD
Monitoring Editor

Andrea L. Marat, PhD
Senior Scientific Editor

Journal of Cell Biology

Reviewer #1 (Comments to the Authors (Required)):

Pellegrino et al investigate the role that peroxisomes play in controlling Mtb infection. They use a series of cutting-edge fluorescence microscopy techniques, in combination with cell biological, genetic, and chemical approaches to argue that peroxisomes are required to restrict the growth of cytologically-localized Mtb in an ESX-1 dependent manner. This study will be of interest to those studying Mtb infection and other intracellular pathogens.

The major conclusions of this paper are:

- 1- Fig 1: Mtb infection transiently induces peroxisome biogenesis and enlargement in an ESX-1-dependent manner.
- 2- Fig 1: Peroxisomes are required to restrict Mtb growth at 24-72hpi, and this restriction is not seen against ESX mutant (Δ RD1). (fig 1)
- 3- Fig 2: Peroxisomal ROS levels are increased in WT-infected cells but not in Δ RD1-infected cells or bystander cells.
- 4- Fig 2: Drug-induced increase in peroxisomal ROS levels inhibits WT Mtb growth, but not RD1 Mtb.
- 5- Fig 3: Restriction of Mtb correlates to higher ROS levels in the cytoplasm.
- 6- Fig 4: The role of peroxisomes is distinct from the NADPH oxidase used to restrict pathogen growth in the phagosome.

The major strengths of this paper are the range of techniques the authors employ to understand how macrophages restrict the growth of an important and difficult to study human pathogen.

The biggest confounding issue is that the statistics used to argue many of the conclusions are confusing and opaque. Specifically, similar-looking distributions are declared significant in one case, but non-significant in another. Importantly, it is unclear what a replicate is in many of the experiments. Does a replicate (N used to perform the statistical test) refer to different cells or different experiments done on different days? Small differences with many N (cells) will be declared significant, whereas the most convincing test relates to reproducibility, i.e. comparing averages across replicates, not individual cells. I understand this is vastly more work, but I do think it will make the conclusions of this paper more rigorous and more convincing.

Major points:

Lines 105 - 130. Nearly all the figures for this entire section are in the supplementals. If this a major point, the figures need to be in the main text.

Fig 1g. and 2a-c. The timing of the events here is confusing, and not entirely consistent with the authors' model. They show that peroxisome number is increased at 24hpi, but not at 48 hpi, but that pathogen restriction in a peroxisome-dependent manner extends out to 72hpi. And they only characterize ROS levels with Pexo_Hyper until 30 hours. What mediates clearance at later stages? What are the peroxisomal ROS levels at later time points in infected cells and bystander cells?

Lines 122-124. "Growth index" not explained in the main text. Also, the statistical tests here are very confusing. Just looking at the underlying distributions, some of the pink bars in the RD1 would appear to be significant. Also, these data don't seem to be consistent with Figure 2G, which doesn't show the growth of the RD1 mutant. In fact, its unclear Mtb area accurately reflects

replication and clearance Some infirmity in the growth metric could strengthen the conclusions, here.

Figure 2B. If these are the same cells followed over time, then a paired t-test seems reasonable. It is also uncertain what the line refers to. It doesn't seem to correlate to the underlying distributions? The differences between B and C could be made more clear in the figure.

Lines 178-180. Its not fair to argue that the compounds don't affect growth of RD1 if there isn't any growth to begin with. Also, to be fair, the first blue point is declared significant, but the authors say there is no difference.

Lines 193-195. I'm very confused about what these data are measuring and what they are arguing. What is the point of the line profile? It doesn't seem to correlate positively or negatively with the orange line.

Figure 3A. There is something weird about the ratiometric imaging. The color scale doesn't seem to either be standardized across the panels, or appear to correspond to the underlying two-color data. For example, Mtb in PEX3^{-/-} is dominated by green, but the ratio seems to be similar to the others. Shouldn't it be very low? Also, suggest not using red/green, but something more color-blind safe.

Figure 3B: Arguing against the author's model, Mtb RD1 appears to have he greatest increase in ROS levels in the PEX-positive macrophages.

Figure 3D, E As in Figure 3C, I'm very confused by these data. What are the zones?

In general, I find Figure 3 to be the weakest in terms of arguing for the authors model, but could be strengthened by better explanation and statistics.

Figure 4D. Some statistics and error bars are desperately needed here.

Minor points:

Lines 90-92. This is confusing. Analyzing doubling positive peroxisomes, and then concluding only one of the markers is there?

Line 114: What's PEX3-Turbo?

Line 62: The authors use the phrase "two-tier" which implies a hierarchy. But they don't explain the hierarchy. Better to say that there are two independent ROS antibacterial mechanisms.

Line 124. Typo, they list Supp "Fig 3E-D" and the D and E should be swapped.

Line 155-156. Typo.

Fig 2g. authors conclude that PPARg mediated ROS production is more relevant given than catalase mediated ROS to pathogen restriction, however, the drugs used to inhibit these pathways don't elevate ROS to the same levels. Could the authors try to boost 3AT (or lessen GW) so that ROS levels are equal, and then test restriction of pathogen growth?

-Fig 3d-e. Suggest labeling which Hyper biosensor is being used in the figure.

-Fig s1f. the CLEM has no quantitation or no real conclusions.

-Fig s2b. Typo ("blue") in the caption.

-Fig s3b. The loading control looks bad, making it hard to trust the conclusion that there is increased catalase expression in the absence of peroxisomes. If they want to conclude that, they will need to do some quantification.

-what does cell area mean in terms of number of bacteria? What does area 500 mean, i.e. units? What is the relationship between cell area and growth index? This is specified eventually in the methods, but might be nice to address in the results or a figure caption.

Reviewer #2 (Comments to the Authors (Required)):

In this study, the authors report the generation of human macrophages that encode fluorescent reporters of peroxisomes or reactive oxygen species (ROS) (or peroxisome knockouts). These cells were used to monitor ROS abundance and the importance of peroxisomes in Mycobacterium tuberculosis (Mtb) infections. The authors conclude that peroxisomes somehow impact the replication of Mtb through ROS. While this conclusion is of interest, the supporting data would need to be strengthened greatly in order for a convincing model to be proposed. The differences in bacterial growth observed and the differences in peroxisome activities, abundance and ROS production are rather modest. This modesty of datasets undermines confidence in the significance of any changes observed. For this reason, enthusiasm is limited.

Specific comments are listed below.

1. Line 36 of the introduction cites review articles to describe prior literature on peroxisomes in host defense. Primary research articles should be cited here, as my comments below require the authors to explain how their work relates to other innate immune activities of these organelles.
2. The studies to analyze peroxisome abundance and morphology in figure 1 need to be amplified, as these organelles, like mitochondria, are not static entities. An increase in peroxisome fission can result in an increase in the appearance of individual organelles without new biogenesis, for example. If peroxisome biogenesis is increased in Mtb infected cells, then the total cellular pool of peroxisomal proteins should increase. The authors are encouraged to examine this possibility by western blotting for Pex14 and other peroxins.
3. Related to point 2, regulators of peroxisome dynamics often regulate mitochondria. The authors are encouraged to perform similar analysis of mitochondria in their infected cells.
4. The increased Mtb replication observed in Pex3 KO cells is minor. It is not clear that a subtle increase in bacterial load per cell (less than 2 fold) is an important aspect of this host-pathogen interaction. Due to the slow rate of Mtb replication in macrophages, it is not clear how the authors can resolve this issue using the time points they have examined. If infections can be reliably monitored over several days (ideally over a week), then the authors are encouraged to extend their analysis.
5. Similar statements to point 4 relate to measurements of peroxisomal ROS in several figures. The changes in ROS observed during infection are rather minor and it is not clear how the authors can validate that such minor changes lead to anti-Mtb activities.
6. Related to prior work on the anti-microbial functions of peroxisomes: How do peroxisome activities or deficiencies relate to the previously reported role of these organelles in interferon and interferon stimulated gene expression? Type 3 interferons have been reported to be induced from these organelles during infection. Does such a phenotype occur in Mtb infected cells? If yes, can the authors subdivide transcriptional changes regulated from peroxisomes from ROS changes, in terms of anti-Mtb activities?

Reviewer #3 (Comments to the Authors (Required)):

In the manuscript "Peroxisomal ROS control cytosolic Mycobacterium tuberculosis replication in human macrophages" by Pellegrino et al., the authors investigate the role of peroxisomes in eliminating the intracellular bacterial pathogen M. tuberculosis (Mtb). Using iPSC-derived human macrophages (iPSDMs) they demonstrate that wild-type but not ESX-1 mutant Mtb triggers peroxisome biogenesis. They show that wild-type Mtb replicates more efficiently in iPSDM lacking peroxisomes but the ESX-1 mutant replication was unimpacted by the loss of peroxisomes. Using ROS probes the authors found that peroxisomes increased ROS level in the cytosol of the Mtb infected iPSDMs. Overall, the authors' data support a model whereby macrophages respond to infection of wild-type/virulent Mtb by generating more peroxisomes leading to the increase of cytosolic ROS which then restricts bacteria that "access the cytosol."

The question investigated by the authors is critical for understanding cell-intrinsic innate immune mechanisms to limit Mtb replication and may provide insights into the development of therapeutics to treat tuberculosis. The authors employ an impressive range of techniques/approaches to explore numerous facets of Mtb-peroxisome biology. Especially impressive are their imaging studies in BSL3 containment, their use of human iPSDMs, the development of novel probes to track ROS in real time, and their execution of extremely well- controlled experiments throughout the manuscript (e.g. fully characterizing KOs before experimentally testing them). That said, there are a couple caveats that should be addressed that would help bolster the authors conclusions drawn in this manuscript.

The authors state that the peroxisome/ROS-mediated mechanism specifically restricts bacteria that access the cytosol. While their data do a great job showing differences in peroxisome biogenesis and killing in wild-type Mtb-infected macrophages, it is unclear whether this is indeed directly due to "cytosolic access." At least, two major innate immune responses can be attributed to the ESX-1 secretion system: the ability to induce type I IFN expression and targeting of damaged Mtb phagosomes to the selective autophagy pathway. I think a couple of additional experiments ruling in or out these responses/processes would bolster the authors' claim that this peroxisome/ROS-mediated mechanism restricts bacterial that access the cytosol.

The authors show that WT infected (and not bystander or ESX-1 infected) macrophages had increase ROS. One explanation of this increased ROS is that type I IFN being generated during WT Mtb infection (and not the ESX-1 mutant) could play a role in generating peroxisomes (required but not sufficient since bystander cells do not have increased ROS). Simply looking at a key readout of peroxisome biogenesis/ROS production during IFN blocking or an IFNAR KO could test the role of IFN in their phenotypes.

It might be challenging, but is there any way to directly show that the cytosolically-exposed bacteria are targeted by ROS/peroxisomes. Do galectin-3/8+ bacteria associate with ROS/peroxisomes?

Point by point reply to reviewers

Reviewer #1

The biggest confounding issue is that the statistics used to argue many of the conclusions are confusing and opaque. Specifically, similar-looking distributions are declared significant in one case, but non-significant in another. Importantly, it is unclear what a replicate is in many of the experiments. Does a replicate (N used to perform the statistical test) refer to different cells or different experiments done on different days? Small differences with many N (cells) will be declared significant, whereas the most convincing test relates to reproducibility, i.e. comparing averages across replicates, not individual cells. I understand this is vastly more work, but I do think it will make the conclusions of this paper more rigorous and more convincing.

We appreciate the reviewer's feedback. We have now modified the legends text to provide clear descriptions of each statistical analysis. In each analysis, we carried out three biological replicates (n = 3) and considered the population number as N, representing the total cells analysed. We pooled the results because we were able to display the same difference as well by analysing the mean between each replicate. An example of this can be seen in Extended Figure 1A. We believe that by including data from all the single cells, we can capture the variability and heterogeneity present in the datasets.

Lines 105 - 130. Nearly all the figures for this entire section are in the supplementals. If this a major point, the figures need to be in the main text.

We agree with the reviewer and now we moved part of the Supplementary figures in the main section (new Figure 1 and 2).

Fig 1g. and 2a-c. The timing of the events here is confusing, and not entirely consistent with the authors' model. They show that peroxisome number is increased at 24hpi, but not at 48 hpi, but that pathogen restriction in a peroxisome-dependent manner extends out to 72hpi. And they only characterize ROS levels with Pexo_Hyper until 30 hours. What mediates clearance at later stages? What are the peroxisomal ROS levels at later time points in infected cells and bystander cells? We thank the reviewer for their thoughtful comment. To study Mtb replication, we typically analyse longer time points (Mtb is a very slow growing pathogen) and lower multiplicities of infection (MOI). Mtb duplicates every 24 hours and prolonged Mtb growth can lead to cell death in macrophages. Therefore, we used a lower MOI of 0.8-1 for replication experiments up to 72-102 hours post-infection (hpi). For all the other experiments, an MOI of 2 was used. Higher MOIs result in macrophages containing more Mtb, increasing the likelihood of damage and localisation of Mtb WT from phagosomes to the cytosol, ultimately leading to increased macrophage necrosis. Therefore, we

typically avoid using higher MOIs (MOI > 2) for analysis at longer time points. However, higher MOIs facilitate a greater number of events available for analysis. Regarding the increase in peroxisomes (observed at MOI: 2), we observed an increase at 24 hpi that was not sustained at 48 hpi. At 24 hpi, Mtb will start to replicate and more than 25% of bacteria in the cytosol (Bernard et al., 2021). We believe that the increase in peroxisomes is not solely due to the number of Mtb, as there is no correlation between PEX14 levels and Mtb replication (new Figure 1G). Instead, it appears to be influenced by replication time and localization of individual Mtb. Furthermore, an increase in changes in peroxisomal shape correlates with functional alterations (Ribeiro et al., 2012). Thus, we focused our analysis on the initial phase of infection (first 30 hpi) to assess any changes in peroxisomal function, particularly in ROS production.

They show that peroxisome number is increased at 24hpi, but not at 48 hpi, but that pathogen restriction in a peroxisome-dependent manner extends out to 72hpi

We believe that the initial phase of restriction (not clearance) at earlier time points, driven by the increase in peroxisomal ROS, plays a crucial role in regulating Mtb growth at later time points. In the absence of peroxisomes, this restriction mechanism is absent, resulting in uncontrolled Mtb growth at early time points. This uncontrolled growth in the early phase has a lasting impact that is reflected at the later time points (see Figure 1G). We have further clarified this in the text.

What mediates clearance at later stages? What are the peroxisomal ROS levels at later time points in infected cells and bystander cells?

We analyzed the peroxisomal ROS ratio after 24 hours post-infection (hpi) (Extended Figure 2B-1). Following 24 hpi, the levels of peroxisomal ROS remained elevated compared to uninfected and Δ RD1 mutant conditions. This sustained elevation of peroxisomal ROS suggests that peroxisomes may enter a phase of degradation (e.g., pexophagy) triggered by the increased ROS levels (Zhang et al., 2015).

Extend Figure 2B-1:

Lines 122-124. "Growth index" not explained in the main text. Also, the statistical tests here are very confusing. Just looking at the underlying distributions, some of the pink bars in the RD1 would appear to be significant. Also, these data don't seem to be consistent with Figure 2G, which doesn't show the growth of the RD1 mutant. In fact, its unclear Mtb area accurately reflects replication and clearance. Some infirmity in the growth metric could strengthen the conclusions, here.

We apologised this was not clear. We added a line in the text explain we are calculating the Mtb growth as growth index, which is calculated using the following formula: $(\text{Mean Mtb area per cell 72 hpi} - \text{Mean Mtb area per cell 2 hpi}) / (\text{Mean Mtb area per cell 2 hpi})$. The growth index enables us to normalize Mtb growth with the time of uptake during the initial 2 hours of infection, ensuring that any variations in growth due to differences in starting time points are accounted for as we have previously shown in other contexts (Santucci et al., 2021; Santucci et al., 2022; Aylan et al., 2023).

Also, these data don't seem to be consistent with Figure 2G, which doesn't show the growth of the RD1 mutant.

This is a good point. To keep consistency, we now changed the graph in 2G to growth index to make it consistent throughout the paper.

Just looking at the underlying distributions, some of the pink bars in the RD1 would appear to be significant.

Macrophages lacking PEX3^{-/-} are unable to restrict the growth of Mtb WT, while the absence of peroxisomes has no effect on phagosomal Mtb Δ RD1. Notably, the growth of Mtb Δ RD1 in PEX3^{-/-} clone 1 (represented in pink) is even more restricted (although not statistically significant) compared to PEX3^{+/+}, further supporting the hypothesis that peroxisomes play a crucial role in restricting Mtb WT (cytosolic) growth but not the Mtb Δ RD1 mutant (phagosomal). We included the data with both clones as they show the same trend.

Figure 2B. If these are the same cells followed over time, then a paired t-test seems reasonable. It is also uncertain what the line refers to. It doesn't seem to correlate to the underlying distributions? The differences between B and C could be made more clear in the figure.

In Figure 2B, we plot the ratio of each analysed cell (represented by dots) along with a trend line and the standard deviation overlaid on the existing plot. This provides an overarching direction of the data. Unfortunately, due to limitations in the current segmentation and tracking system, we were unable to track individual cells over time. However, we are actively working on improving this system that requires better temporal resolution. This is proving to be challenging with the current probes. Nonetheless, we aimed to showcase the mean ratio at the single-cell level to highlight the heterogeneity observed during infection. We added a statistical analysis for the last two timepoint (27 hpi and 30 hpi) and significance was determined by two-way ANOVA with Tukey's post-doc test. See extended data 2B-2 for statistical comparison.

The differences between B and C could be made more clear in the figure.

We have made changes to Figure 2C to improve the visual representation of the data.

Extend Figure 2B-2:

Statistics: two-way Anova, Tukey's multiple comparisons test

Its not fair to argue that the compounds don't affect growth of RD1 if there isn't any growth to begin. We understand and agree with this point. Mtb Δ RD1 is an attenuated strain and usually the growth is not obvious as for the Mtb WT strain, but this mutant can grow (as shown in the images in Fig. 2C). We changed the graph 2G to reflect this better.

Also, to be fair, the first blue point is declared significant, but the authors say there is no difference. We agree with the reviewer, and we have now rephrased the sentence in the main text.

Lines 193-195. I'm very confused about what these data are measuring and what they are arguing. What is the point of the line profile? It doesn't seem to correlate positively or negatively with the orange line. We apologise for the lack of clarity in this figure. We use the profile line to show that nearby the Mtb area the total level of ROS was decreasing in PEX3^{-/-} macrophages infected with Mtb WT. By analysing the orange line, indicating the ratio of the hyper sensor, we show that the ratio decreases in the macrophages lacking peroxisomes. To make it clearer we used the same scale for both conditions and clarified further this point in the text.

Figure 3A. There is something weird about the ratiometric imaging. The color scale doesn't seem to either be standardized across the panels or appear to correspond to the underlying two-color data. For example, Mtb in PEX3^{-/-} is dominated by green, but the ratio seems to be similar to the others. Shouldn't it be very low? Also, suggest not using red/green, but something more color-blind safe.

We agree. The ratio has been checked in all the images and the Mtb WT PEX3^{-/-} image has been changed with more representative image. See new figure 3A. Unfortunately, due to time constraints, we were unable to make all the figures colour-blind friendly.

Figure 3B: Arguing against the author's model, Mtb RD1 appears to have the greatest increase in ROS levels in the PEX-positive macrophages.

We agree with the reviewer. We see an increase in cytosolic ROS in Mtb Δ RD1 infected PEX3^{+/+} macrophages (although is not significant) and we believe this is due to the ROS leaking from the phagosomes ([https://www.jbc.org/article/S0021-9258\(20\)76819-5/fulltext](https://www.jbc.org/article/S0021-9258(20)76819-5/fulltext); reviewed in <https://www.hindawi.com/journals/tswj/2011/741046/>). The striking point is that the slightly increase in ROS in the Mtb WT infected PEX3^{+/+} macrophages completely decrease in the infected PEX3^{-/-} macrophages but this is not reflected in the macrophages infected with Mtb Δ RD1. These data suggest the increase of cytosolic ROS in Mtb Δ RD1 infected macrophages it is not produced from peroxisomes but from other source, such as NADPH oxidase present on phagosomes.

Figure 3D, E As in Figure 3C, I'm very confused by these data. What are the zones?

We agree with the reviewer that the way the data is shown is confusing. For better understanding, we decided to show the Endo_ROS ratio around Mtb. We modified figure 3E and the main text accordingly.

In general, I find Figure 3 to be the weakest in terms of arguing for the authors model but could be strengthened by better explanation and statistics.

We now explain better and reviewed modified the statistics accordingly. We thank the reviewer for bringing this point. We believe that the modifications we made during this review have effectively strengthened the message.

Figure 4D. Some statistics and error bars are desperately needed here.

We agree with the reviewer. We added new statistics analysis in the figure 4D for the last timepoint.

Lines 90-92. This is confusing. Analyzing doubling positive peroxisomes, and then concluding only one of the markers is there?

We have made modifications in the main text.

Line 114: What's PEX3-Turbo?

We have now indicated that in the main text.

Line 62: The authors use the phrase "two-tier" which implies a hierarchy. But they don't explain the hierarchy. Better to say that there are two independent ROS antibacterial mechanisms.

This is a good point. We have now clarified this point in the discussion.

Line 124. Typo, they list Supp "Fig 3E-D" and the D and E should be swapped.

We now corrected it.

Line 155-156. Typo.

We now corrected it.

Fig 2g. authors conclude that PPAR γ mediated ROS production is more relevant given than catalase mediated ROS to pathogen restriction, however, the drugs used to inhibit these pathways don't elevate ROS to the same levels. Could the authors try to boost 3AT (or lessen GW) so that ROS levels are equal, and then test restriction of pathogen growth?

We appreciate the reviewer's point. In our study, we intentionally chose a concentration of 3-AT that targets catalase activity in macrophages (see Extended Figure S4), rather than focusing solely on ROS

production. We do not anticipate that increasing the 3-AT concentration would lead to a significant rise in peroxisomal ROS levels, as catalase activity is already fully inhibited. Consequently, our conclusion argues that ROS production plays a crucial role in this antibacterial mechanism, emphasizing its significance over ROS degradation.

Extended Figure S4:

Fig 3d-e. Suggest labeling which Hyper biosensor is being used in the figure. We add “Ratio Endo Hyper” in Figure 3E.

Fig s1f. the CLEM has no quantitation or no real conclusions.

We acknowledge that this experiment is purely qualitative, and we are not drawing any conclusive findings from it. We now removed it from the manuscript.

Fig s2b. Typo ("blue") in the caption.

We have made modifications to the caption.

Fig s3b. The loading control looks bad, making it hard to trust the conclusion that there is increased catalase expression in the absence of peroxisomes. If they want to conclude that, they will need to do some quantification.

We agree with the reviewer about this claim, and we have now quantified the Catalase, see Extended Figure FS3B-1 now added as Fig. S2C.

Extended Figure FS3B-1:

What does cell area mean in terms of number of bacteria? What does area 500 mean, i.e. units? What is the relationship between cell area and growth index? This is specified eventually in the methods, but might be nice to address in the results or a figure caption.

"Mtb area per cell" represents the area of Mtb within a single cell, expressed in square pixels (px²). We add pixel square in the caption of the supplementary figures. We prefer using “area” instead of “number”

of individual bacteria due to segmentation limitations. Counting them may result in erroneously including multiple bacteria when they are in proximity. Analysing the area overcomes this challenge. In terms of the relationship with Mtb per cells and growth index, it is that we add the normalization step with the 2hpi time point, as explained in the material and methods. We have now briefly commented on this in the results.

Reviewer #2

The supporting data would need to be strengthened greatly in order for a convincing model to be proposed. The differences in bacterial growth observed and the differences in peroxisome activities, abundance and ROS production are rather modest. This modesty of datasets undermines confidence in the significance of any changes observed. For this reason, enthusiasm is limited.

We understand the reviewer concerns and addressed these points below.

Line 36 of the introduction cites review articles to describe prior literature on peroxisomes in host defense. Primary research articles should be cited here, as my comments below require the authors to explain how their work relates to other innate immune activities of these organelles.

We appreciate the review and share the same concern. To address this, we have incorporated new references in the introduction. There is emerging literature showing that peroxisomes play a crucial role in the immune response and antibacterial functions. The work presented here has revealed another potential antibacterial function focus on the restriction of cytosolic Mtb through peroxisomal ROS. However, it will be important to conduct further studies with other cytosolic pathogens, to unravel the underlying mechanism and explore any transcriptional changes linked to this significant discovery.

The studies to analyze peroxisome abundance and morphology in figure 1 need to be amplified, as these organelles, like mitochondria, are not static entities. An increase in peroxisome fission can result in an increase in the appearance of individual organelles without new biogenesis, for example. If peroxisome biogenesis is increased in Mtb infected cells, then the total cellular pool of peroxisomal proteins should increase. The authors are encouraged to examine this possibility by western blotting for Pex14 and other peroxins.

We agree it would be important to analyse other organelle and their dynamics. While we acknowledge the reviewer's perspective regarding the involvement of other organelles in Mtb restriction, our primary focus in this study was to explore the new function and role of peroxisomes during infection. Analysing mitochondrial dynamics in this context is beyond the scope (and the limited space) of this study.

An increase in peroxisome fission can result in an increase in the appearance of individual organelles without new biogenesis, for example. If peroxisome biogenesis is increased in Mtb infected cells, then the total cellular pool of peroxisomal proteins should increase.

At 24 hours post-infection (hpi) with Mtb, only approximately 30% of cells will be infected, and we believe that this dynamic will go undetected through western blot analysis. Therefore, we opted for Immunofluorescence analysis. Nonetheless, we conducted western blot analysis for PEX14, but as well ACOX1, HSD17B4 and Catalase proteins during infection. Interestingly, we observed an upregulation of these peroxisomal protein primarily in Mtb WT infected cells compared to uninfected cells and Mtb Δ RD1 (see Extended Figure 1) now added as Fig 1I.

Extended Figure 1:

Related to point 2, regulators of peroxisome dynamics often regulate mitochondria. The authors are encouraged to perform similar analysis of mitochondria in their infected cells.

In a previous study (Bussi et al., 2022) and based on internal data, we demonstrated alterations in mitochondrial function during infection with Mtb. While we acknowledge the potential connection between these organelles, our current study specifically emphasizes the role of peroxisomes during infection. Therefore, we chose not to extensively investigate mitochondrial analysis in this particular study.

The increased Mtb replication observed in Pex3 KO cells is minor. It is not clear that a subtle increase in bacterial load per cell (less than 2fold) is an important aspect of this host-pathogen interaction. Due to the slow rate of Mtb replication in macrophages, it is not clear how the authors can resolve this issue using the time points they have examined. If infections can be reliably monitored over several days (ideally over a week), then the authors are encouraged to extend their analysis.

We understand the concern of the reviewer. At the same time, for a slow-growing bacteria, a two-fold increase is already an important phenotype to take in consideration. Unfortunately, extending the infection time with induced pluripotent stem cell-derived macrophages (iPSDM) is challenging because prolonged exposure to Mtb leads to cell death due to bacterial replication (Aylan, Bernard, Pellegrino et al., 2023). To address this limitation, we employed an alternative *in vitro* model using human monocyte-derived macrophages (HMDM), which allowed us to extend the infection duration to later timepoints (up to ~100 hpi). In this model, we observed a consistent increase in Mtb wild-type growth in the PEX3^{nf} condition over time (refer to Figure 4D).

Similar statements to point 4 relate to measurements of peroxisomal ROS in several figures. The changes in ROS observed during infection are rather minor and it is not clear how the authors can validate that such minor changes lead to anti-Mtb activities.

We understand the concern of the reviewer and are grateful this concern was raised. We believe that the increase of peroxisomal ROS, although minor, is critical for Mtb WT restriction. This is supported by the Hyper-Cyto experiment (Fig 3A-C). In our system, we think there are no large increases of H₂O₂ because: 1. The production of H₂O₂ is a very dynamic event and the H₂O₂ formed will be release from peroxisomes by porin or other unknown mechanisms (Lismont et al, 2019). 2. An increase of peroxisomal H₂O₂ will bring to an increase of peroxisomal peroxidation and so a decrease of peroxisomal structures. 3. A high H₂O₂ concentration will lead to cell death. Our argument is that the cell has this antibacterial control but that will only function if the cell is alive.

Related to prior work on the anti-microbial functions of peroxisomes: How do peroxisome activities or deficiencies relate to the previously reported role of these organelles in interferon and interferon stimulated gene expression? Type 3 interferons have been reported to be induced from these organelles during infection. Does such a phenotype occur in Mtb infected cells? If yes, can the authors subdivide transcriptional changes regulated from peroxisomes from ROS changes, in terms of anti-Mtb activities? Interferon Lambda (type III) has been associated to infection and its increased expression correlate with an increase of peroxisomal number (Odendall et al., 2014). Thus, we checked expression of IFN during infection with qPCR in PEX3 +/+ and PEX3 -/- (see **Extended Data 1A**). From the qPCR data, we

observed an increase of most all the IFN during only Mtb WT infection, but this increase it is sustained as well in PEX3 ^{-/-} macrophages, suggesting that the absence of peroxisome it is not influencing the expression of IFN Lambda.

Reviewer #3

The authors state that the peroxisome/ROS-mediated mechanism specifically restricts bacteria that access the cytosol. While their data do a great job showing differences in peroxisome biogenesis and killing in wild-type Mtb-infected macrophages, it is unclear whether this is indeed directly due to "cytosolic access." At least, two major innate immune responses can be attributed to the ESX-1 secretion system: the ability to induce type I IFN expression and targeting of damaged Mtb phagosomes to the selective autophagy pathway. I think a couple of additional experiments ruling in or out these responses/processes would bolster the authors' claim that this peroxisome/ROS-mediated mechanism restricts bacterial that access the cytosol.

We appreciate the suggestion from the reviewer. For this reason, we decide to analyse the IFN response during infection (see Extended Data 1), but we conclude that the IFN response, although if present during infection, it is not causing the phenotype we are describing in this manuscript.

The authors show that WT infected (and not bystander or ESX-1 infected) macrophages had increase ROS. One explanation of this increased ROS is that type I IFN being generated during WT Mtb infection (and not the ESX-1 mutant) could play a role in generating peroxisomes (required but not sufficient since bystander cells do not have increased ROS). Simply looking at a key readout of peroxisome biogenesis/ROS production during IFN blocking or an IFNAR KO could test the role of IFN in their phenotypes.

We appreciate the suggestion from the reviewer. For this reason, we decided to analyse the IFN response during infection and we used three different drugs: Tofacitinib (Jak inhibitor), Anifrolumab (Anti-IFNAR) and Sifalimumab (Anti-IFN α) during infection (see Extended Data 1B). From this result we concluded that Tofacitinib is a good drug to inhibit the activation of the STAT signalling. Then we tested if these drugs were modulating peroxisomal ROS (see Extended Data 1C). Although only Tofacitinib were able to increase peroxisomal ROS, we are not sure what is the mechanism behind. Then we tested these drugs during infection with Mtb WT and Mtb Δ RD1 (see Extended Data 1D). From this experiment, only Tofacitinib were able to restrict Mtb WT (three independent replicates), again supporting the hypothesis that peroxisomal ROS it is important to control Mtb WT infection. In addition, we confirm that this was only a host phenotype, because the drugs were not inhibiting Mtb *in vitro*, see Extended Data 1E. Moreover, we calculate if the number of peroxisomes were changing during infection with Mtb WT in presence of drugs (see Extended Data 1F), but we did not observe any changes, supporting the idea that probably the mechanism we are describing is independent of the STAT pathway.

Extended Data 1- IFN modulation during Mtb infection in iPSDM.

(A), Relative expression analysis of IFN protein by qPCR at 24 hpi in Uninfected, Mtb WT and Mtb Δ RD1. Expression levels were normalized to the housekeeping gene GADPH using the $2^{-\Delta\Delta Ct}$ method. (B), Representative Western Blot of STAT1, p-STAT1 in iPSDM during infection (Uninfected, Mtb WT and Mtb Δ RD1) untreated or treated with INF- α , Tocifitinib (TOCI), Anifrolumab (ANF), Sifalimumab (SIL). (C), Pezo Hyper Ratio analysis of iPSDM untreated or treated with INF- α , Tocifitinib (TOCI), Anifrolumab (ANF), Sifalimumab (SIL) for 24 hours. (D), Fold change of Mtb WT (left) and Mtb Δ RD1 (right) in iPSDM untreated or treated with INF- α , Tocifitinib (TOCI), Anifrolumab (ANF), Sifalimumab (SIL). (E), *In vitro* analysis of Mtb growth untreated or treated with INF- α , Tocifitinib (TOCI), Anifrolumab (ANF), Sifalimumab (SIL). (F), Quantification of PEX14 puncta per cells at 24 hpi of infection in iPSDM untreated or treated with INF- α , Tocifitinib (TOCI), Anifrolumab (ANF), Sifalimumab (SIL).

It might be challenging but is there any way to directly show that the cytosolically-exposed bacteria are targeted by ROS/peroxisomes. Do galectin-3/8+ bacteria associate with ROS/peroxisomes?

We appreciate the reviewer's understanding of the challenges we encountered during our study. While we did observe colocalization between the EGFP_PTS1 and Gal3 markers in our internal data, we determined that it was not informative for the specific objective of our paper, which was to investigate peroxisomal ROS targeting of cytosolic Mtb. To analyze peroxisomal ROS, we specifically utilized the Hyper reporter, which is compatible with live-cell imaging only. Unfortunately, this restricted our ability to perform staining for other markers such as Gal3. Additionally, due to technical limitations, we were unable to express both the Hyper reporter and Gal3 simultaneously in our experimental system. We sincerely appreciate the reviewer's feedback and suggestions, and we will consider them for future research directions or alternative methodologies.

August 23, 2023

RE: JCB Manuscript #202303066R

Dr. Maximiliano G Gutierrez
The Francis Crick Institute
1 Midland Road
London NW1 1AT
United Kingdom

Dear Dr. Gutierrez:

Thank you for submitting your revised manuscript entitled "Peroxisomal ROS control cytosolic Mycobacterium tuberculosis replication in human macrophages". We would be happy to publish your paper in JCB pending final revisions necessary to meet our formatting guidelines (see details below). In your final revision, please be sure to address reviewer #2's final minor concerns.

A. MANUSCRIPT ORGANIZATION AND FORMATTING:

1) Text limits: Character count for Reports is < 20,000, not including spaces. Count includes abstract, introduction, * combined results and discussion, and acknowledgments. Count does not include title page, figure legends, materials and methods, references, tables, or supplemental legends.

* You may exceed the character count slightly as necessary to address the final comments.

2) Figures limits: Reports may have up to 5 main text figures.

3) * Figure formatting: Scale bars must be present on all microscopy images, including inset magnifications. Molecular weight or nucleic acid size markers must be included on all gel electrophoresis. In order to accommodate readers with red-green color blindness, we highly suggest you avoid red-green when possible. *

4) Statistical analysis: Error bars on graphic representations of numerical data must be clearly described in the figure legend. The number of independent data points (n) represented in a graph must be indicated in the legend. Statistical methods should be explained in full in the materials and methods. For figures presenting pooled data the statistical measure should be defined in the figure legends. Please also be sure to indicate the statistical tests used in each of your experiments (either in the figure legend itself or in a separate methods section) as well as the parameters of the test (for example, if you ran a t-test, please indicate if it was one- or two-sided, etc.). Also, if you used parametric tests, please indicate if the data distribution was tested for normality (and if so, how). If not, you must state something to the effect that "Data distribution was assumed to be normal but this was not formally tested."

5) Abstract and title: The abstract should be no longer than 160 words and should communicate the significance of the paper for a general audience. The title should be less than 100 characters including spaces. Make the title concise but accessible to a general readership. We request that titles do not contain species names except if essential.

6) Materials and methods: Should be comprehensive and not simply reference a previous publication for details on how an experiment was performed. Please provide full descriptions in the text for readers who may not have access to referenced manuscripts.

7) Please be sure to provide the sequences for all of your primers/oligos and RNAi constructs in the materials and methods. You must also indicate in the methods the source, species, and catalog numbers (where appropriate) for all of your antibodies. Please also indicate the acquisition and quantification methods for immunoblotting/western blots.

8) Microscope image acquisition: The following information must be provided about the acquisition and processing of images:

- a. Make and model of microscope
- b. Type, magnification, and numerical aperture of the objective lenses
- c. Temperature
- d. Imaging medium
- e. Fluorochromes

f. Camera make and model

g. Acquisition software

h. Any software used for image processing subsequent to data acquisition. Please include details and types of operations involved (e.g., type of deconvolution, 3D reconstitutions, surface or volume rendering, gamma adjustments, etc.).

10) Supplemental materials: There are strict limits on the allowable amount of supplemental data. Reports may have up to 3 supplemental figures. Please also note that tables, like figures, should be provided as individual, editable files. A summary of all supplemental material should appear at the end of the Materials and methods section.

13) ORCID IDs: ORCID IDs are unique identifiers allowing researchers to create a record of their various scholarly contributions in a single place. Please note that ORCID IDs are now *required* for all authors. At resubmission of your final files, please be sure to provide your ORCID ID and those of all co-authors.

Please note that JCB now requires authors to submit Source Data used to generate figures containing gels and Western blots with all revised manuscripts. This Source Data consists of fully uncropped and unprocessed images for each gel/blot displayed in the main and supplemental figures. Since your paper includes cropped gel and/or blot images, please be sure to provide one Source Data file for each figure that contains gels and/or blots along with your revised manuscript files. File names for Source Data figures should be alphanumeric without any spaces or special characters (i.e., SourceDataF#, where F# refers to the associated main figure number or SourceDataFS# for those associated with Supplementary figures). The lanes of the gels/blots should be labeled as they are in the associated figure, the place where cropping was applied should be marked (with a box), and molecular weight/size standards should be labeled wherever possible.

Journal of Cell Biology now requires a data availability statement for all research article submissions. These statements will be published in the article directly above the Acknowledgments. The statement should address all data underlying the research presented in the manuscript. Please visit the JCB instructions for authors for guidelines and examples of statements at (<https://rupress.org/jcb/pages/editorial-policies#data-availability-statement>).

B. FINAL FILES:

**It is JCB policy that if requested, original data images must be made available to the editors. Failure to provide original images

upon request will result in unavoidable delays in publication. Please ensure that you have access to all original data images prior to final submission.**

Thank you for this interesting contribution, we look forward to publishing your paper in Journal of Cell Biology.

Sincerely,

Craig Roy, PhD
Monitoring Editor

Andrea L. Marat, PhD
Senior Scientific Editor

Journal of Cell Biology

Reviewer #1 (Comments to the Authors (Required)):

The authors have significantly improved the manuscript through revision and response to reviewers comments.

Reviewer #2 (Comments to the Authors (Required)):

In this revised manuscript, the authors have addressed my comments. New data is presented by western analysis to suggest that infections lead to an increase in peroxisome abundance. This analysis bolsters the microscopic studies present in the initial manuscript.

I remain concerned about the modest nature of the phenotypes observed, in terms of bacterial replication and ROS production. This modesty could be discussed more directly in the abstract and discussion, in order to ensure that the reader is not oversold on the conclusions offered.

Reviewer #3 (Comments to the Authors (Required)):

The authors have more than adequately addressed my critiques/concerns. Nice work!!!